# Discriminator-Weighted Offline Imitation Learning from Suboptimal Demonstrations

## Abstract

We study the problem of offline Imitation Learning (IL) where an agent aims to learn an optimal expert behavior policy without additional online environment interactions. Instead, the agent is provided with a static offline dataset of state-action-next state transition triples from both optimal and non-optimal behaviors. This strictly offline imitation learning problem arises in many real-world problems, where environment interactions and expert annotations are costly. Prior works that address the problem either require that expert data occupies the majority proportion of the offline dataset, or need to learn a reward function and perform offline reinforcement learning (RL) based on the learned reward function. In this paper, we propose an imitation learning algorithm to address the problem without additional steps of reward learning and offline RL training for the case when demonstrations containing large proportion of suboptimal data. Built upon behavioral cloning (BC), we introduce an additional discriminator to distinguish expert and non-expert data, we propose a cooperation strategy to boost the performance of both tasks, this will result in a new policy learning objective and surprisingly, we find its equivalence to a generalized BC objective, where the outputs of discriminator serve as the weights of the BC loss function. Experimental results show that our proposed algorithm achieves higher returns and faster training speed compared to baseline algorithms.

## 1 Introduction

The recent success of reinforcement learning (RL) in many domains showcases the great potential of applying this family of learning methods to real-world applications. A key prerequisite for RL is to design a reward function that specifies what kind of agent behavior is preferred. However, in many real-world applications, designing a reward function is prohibitively difficult (Ng et al., 1999; Irpan, 2018). By contrast, imitation learning (IL) provides a much easier way to leverage the reward function implicitly from the collected demonstrations and has achieved great success in many sequential decision making problems (Pomerleau, 1989; Ng et al., 2000; Ho & Ermon, 2016).

However, popular IL methods such as behavioral cloning (BC) and generative adversarial imitation learning (GAIL) (Ho & Ermon, 2016), assume the expert demonstration is optimal. Unfortunately, it is often difficult to obtain optimal demonstrations for many real-world tasks, because human experts often make mistakes due to various reasons, such as the difficulty of the task, partial observability of the environment, or the presence of distraction. Given such noisy expert demonstrations, which contain records of both optimal and non-optimal behaviors, BC and GAIL all fail to imitate the optimal policy (Wu et al., 2019b; Ma, 2020). Current methods that deal with suboptimal demonstrations either require additional labels, which can be done explicitly by annotating each demonstration with confidence scores by human experts (Wu et al., 2019b), or implicitly by ranking noisy demonstrations according to their relative performance through interacting with the environment (Brown et al., 2019; 2020). However, human annotation and environment interaction are laborious and expensive in real-world settings, such as in medicine, healthcare, and industrial processes.

In this work, we investigate a pure offline learning setting where the agent has access to neither the expert nor the environment for additional information. The agent, instead, has only access to a small pre-collected dataset of state-action-next state transition triples sampled from the expert and a large batch offline dataset sampled from one or multiple behavior policies that could be highly sub-optimal. Prior works that address the problem are based on variants of BC or inverse RL, Sasaki

& Yamashina (2021) reuse another policy learned by BC as the weight of original BC objective, however, this requires that expert data occupy the majority proportion of the offline dataset, otherwise the policy will be misguided to imitate the suboptimal data. Zolna et al. (2020a) first learns a reward function that prioritizes expert data over others and then performs offline RL based on this reward function. This algorithm is extremely expensive to run, requiring solving offline RL in an inner loop, which itself is a challenging problem and prone to training instability (Kumar et al., 2019) and hyperparameter sensitivity (Wu et al., 2019a).

In this paper, we propose an offline imitation learning algorithm to learn from demonstrations that (perhaps) contain a large proportion of suboptimal data without additional steps of reward learning and offline RL training. Built upon the task of behavioral cloning (BC), we introduce an additional task to learn a discriminator to distinguish expert and non-expert data, we propose a cooperation strategy to boost the performance of both tasks. This results in a new policy learning objective and surprisingly, we find its equivalence to a generalized BC objective, where the outputs of the discriminator serve as the weights of the BC loss function. We thus term our resulting algorithm Discriminator-Weighted Behavioral Cloning (DWBC). Experimental results show that DWBC achieves higher returns and faster training speed compared to baseline algorithms, under different scenarios.

## 2 PRELIMINARY

### 2.1 PROBLEM SETTING

We consider the standard fully observed Markov Decision Process (MDP) setting (Sutton et al., 1998), $\mathcal{M} = \{\mathcal{S}, \mathcal{A}, P, r, \gamma, d_0\}$, where $\mathcal{S}$ is the state space, $\mathcal{A}$ is the action space, $P : \mathcal{S} \times \mathcal{A} \to \Delta(\mathcal{S})$ is the MDP's transition probability, $r$ is the reward function, $\gamma \in [0, 1)$ is the discount factor for future reward and $d_0$ is the initial distribution. A policy $\pi : \mathcal{S} \to \Delta(\mathcal{A})$ maps from state to distribution over actions. We denote $d^\pi \in \Delta(\mathcal{S} \times \mathcal{A})$ as the discounted state-action distribution of $\pi$ under transition kernel $P$, that is, $d^\pi = (1 - \gamma) \sum_{t=0}^\infty \gamma^t d_t^\pi$, where $d_t^\pi \in \Delta(\mathcal{S} \times \mathcal{A})$ is the distribution of $\left(s^{(t)}, a^{(t)}\right)$ under $\pi$ at step $t$. Following the standard IL setting, the ground truth reward function $r$ is unknown. Instead, we have the demonstrations by the expert specified by $\pi_e : \mathcal{S} \to \Delta(\mathcal{A})$ (potentially stochastic and not necessarily optimal). Concretely, we have an expert dataset in the form of i.i.d tuples $\mathcal{D}_e = \{s_i, a_i, s_i'\}_{i=1}^{n_e}$ where $(s, a)$ is sampled from distribution $d^{\pi_e}$ and $s'$ is sampled from $P(s, a)$.

In our problem setting, we also have an offline static dataset consisting of i.i.d tuples $\mathcal{D}_o = \{s_i, a_i, s_i'\}_{i=1}^{n_o}$ s.t. $(s, a) \sim \rho(s, a), s' \sim P(s, a)$, where $\rho \in \Delta(\mathcal{S} \times \mathcal{A})$ is an offline state-action distribution resulting from some other behavior policies. Note that these behavior policies could be much worse than the expert $\pi_e$. Our goal is to only leverage the offline batch data $\mathcal{D}_b = \mathcal{D}_e \cup \mathcal{D}_o$ to learn an optimal policy $\pi$ with regard to optimizing the ground truth reward $r$, without any interaction with the environment or expert.

### 2.2 A GENERALIZED BEHAVIORAL CLONING OBJECTIVE

In order to discard low-quality demonstrations and only clone the best behavior available, previous works (Sasaki & Yamashina, 2021; Zolna et al., 2020a) consider a generalized behavioral cloning objective to imitate demonstrations unequally, that is,

$$\min_\pi \mathbb{E}_{(s,a) \sim \mathcal{D}_b} \left[ -\log \pi(a|s) \cdot f(s, a) \right] \tag{1}$$

where $f : \mathcal{S} \times \mathcal{A} \to [0, 1]$ denotes an arbitrary weight function.

- If $f(s, a) = 1$ for $\forall (s, a) \in \mathcal{S} \times \mathcal{A}$, the objective (1) corresponds to the vanilla BC objective.
- If $f(s, a) = \pi'(a|s)$, where $\pi'$ is an old policy which was previously optimized with $\mathcal{D}_b$, the objective (1) corresponds to the objective of Behavioral Cloning from Noisy Demonstrations (Sasaki & Yamashina, 2021). Since $\sum_a \pi'(a|s) = 1$ for $\forall s \in \mathcal{S}$ is satisfied, $\pi'(a|s)$ can be interpreted as the weights for the weighted action sampling.
- If $f(s, a) = \mathbb{1}\left[A^\pi(s, a)\right]$, where $\mathbb{1}$ is the indicator function which creates a boolean mask that eliminates samples which are thought to be worse than the current policy, the objective (1) corresponds to the objective of Offline Reinforced Imitation Learning (Zolna et al., 2020a).

The objective (1) can also be deemed as the objective of Soft Q Imitation Learning (Reddy et al., 2019) with $f(s, a) = 1$ for $(s, a) \in \mathcal{D}_e$ and $f(s, a) = 0$ for $(s, a) \in \mathcal{D}_o$ in online IL literature; or the objective of off-policy actor-critic (Off-PAC) algorithm (Degris et al., 2012) with $f(s, a) = Q^\pi(s, a) \cdot \pi(a|s)/\pi_b(a|s)$ in online RL literature.

## 3 DISCRIMINATOR-WEIGHTED BEHAVIORAL CLONING

We now continue to describe our approach for offline imitation learning from demonstrations that (perhaps) contain large-proportional suboptimal data without additional steps of reward learning and offline RL training. Built upon the task of BC, we introduce an additional task to learn a discriminator to distinguish expert and non-expert data, we propose a cooperation strategy to boost the performance of both tasks, this will result in a new generalized BC objective. We then provide the interpretation of weights in our generalized BC objective, this gives the intuition about why our method can work.

### 3.1 LEARN THE POLICY AND DISCRIMINATOR SEPARATELY

It is obvious that we can avoid the negative impact of suboptimal demonstrations presented in $\mathcal{D}_o$ by only imitating $\mathcal{D}_e$, which can be written as

$$\min_{\pi} \mathbb{E}_{(s,a)\sim\mathcal{D}_e} [-\log \pi(a|s)] \tag{2}$$

We call the task of learning a policy using objective (2) as the BC task. The drawback of BC task is that the learned policy may not be able to generalize due to the potential limited size and state coverage of $\mathcal{D}_e$. It does not fully utilize the information from $\mathcal{D}_o$. If we can select those high-reward transitions from $\mathcal{D}_o$ and combine them with $\mathcal{D}_e$, we are expected to get a better policy.

Now let's consider another different task, which aims to learn a discriminator by contrasting expert and non-expert transitions, given by

$$\min_{d} \mathbb{E}_{(s,a)\sim\mathcal{D}_e} [-\log d(s, a)] + \mathbb{E}_{(s,a)\sim\mathcal{D}_o} [-\log(1 - d(s, a))] \tag{3}$$

We call this the discriminating task, which is similar to how the discriminator is trained in GAIL (Ho & Ermon, 2016) and GAN (Goodfellow et al., 2014a), except that the second term is sampled from a fixed dataset instead of new samples obtained from interacting with the environment.

However, optimizing objective (3) will make the discriminator learned to assign 1 to all transitions from $\mathcal{D}_e$ and 0 to all transitions from $\mathcal{D}_o$. This limiting behavior is unsatisfactory because $\mathcal{D}_o$ can contain some successful (high-reward) transitions. This bears similarity to the positive-unlabeled classification problem (Elkan & Noto, 2008), where both positive and negative samples exist in the unlabeled data.

To solve this problem, we adopt the approach from positive-unlabeled (PU) learning (Du Plessis et al., 2015; Xu & Denil, 2019; Zolna et al., 2020b). The main idea is to re-weight different losses for positive and unlabeled data, in order to obtain an estimate of model loss on negative samples that is not directly available. Applying PU learning to objective (3) yields the following objective:

$$\min_{d} \eta \mathbb{E}_{(s,a)\sim\mathcal{D}_e} [-\log d(s, a)] + \mathbb{E}_{(s,a)\sim\mathcal{D}_o} [-\log(1 - d(s, a))] - \eta \mathbb{E}_{(s,a)\sim\mathcal{D}_e} [-\log(1 - d(s, a))] \tag{4}$$

where $\eta$ is a hyperparameter, corresponds to the proportion of positive samples to unlabeled samples. Intuitively, the second term in (4) could make $d(s, a)$ of state-action pairs from $\mathcal{D}_e$ become 0 if similar state-action pairs are included in $\mathcal{D}_o$, and the third term in (4) balances the impact of the second term, i.e., avoids $d(s, a)$ of state-action pairs from $\mathcal{D}_e$ becoming 0.

To summarize, the BC task aims to imitate the expert behavior from $\mathcal{D}_e$, but ignores the valuable information in $\mathcal{D}_o$; the discriminating task aims to contrast expert and non-expert transitions from $\mathcal{D}_e$ and $\mathcal{D}_o$, by only using state-action information as input. Both tasks lack enough information to improve their own performance, which, however, can be obtained from the other task. It then seems natural to find a scheme to incorporate the policy into the training of the discriminator and effectively use the discriminator to help the training of the policy.

### 3.2 LEARN THE POLICY AND DISCRIMINATOR COOPERATIVELY

We propose a cooperation strategy to boost the performance of both tasks. To boost the performance of the discriminating task, we add the imitation information from $\pi$ to the input of the discriminator, yielding the following discriminator learning objective $\mathcal{L}_d$ to be minimized as:

$$\min_d \mathcal{L}_d = \min_d \eta \mathop{\mathbb{E}}_{(s,a)\sim\mathcal{D}_e} [-\log d(s, a, \log \pi(a|s))] +$$
$$\mathop{\mathbb{E}}_{(s,a)\sim\mathcal{D}_o} [-\log(1 - d(s, a, \log \pi(a|s)))] - \eta \mathop{\mathbb{E}}_{(s,a)\sim\mathcal{D}_e} [-\log(1 - d(s, a, \log \pi(a|s)))] \quad (5)$$

Supposed $\pi$ is learned to be optimal, i.e., assigns large probabilities to expert actions in expert states, the discriminator will receive additional learning signal. It will be easier for the discriminator to contrast expert and non-expert transitions in $\mathcal{D}_o$, as $\log \pi(a|s)$ will be large if $(s, a)$ are from expert behaviors and small if $(s, a)$ are from non-expert behaviors. Without this information from $\pi$, the discriminator is much harder to learn by only using information from $(s, a)$.

Now let's dive deeper into objective (5). As $\pi$ now appears in the input of the discriminator $d$, hence both $d$ and its loss $\mathcal{L}_d$ become functionals of $\pi$. We are interested to see how does the imitation information from $\log \pi$ affect $\mathcal{L}_d$, and further impact $d$. In other words, given the current $d$, we want to explore how to change the behavior of $\pi$ such that $d$ can be better learned. To achieve this goal, we define a functional $J(\pi)$ for $\mathcal{L}_d$ with the function $\pi$ as the variable, and fix the parameter $\theta_d$ of $d$ to exclude its own influence. Note that in the functional $J(\pi)$, $d$ is now fixed and its parameter $\theta_d$ will no longer be considered as a variable. $J(\pi)$ can be formally define as the following integral form to further eliminate the effect of changes in $s, a$ on $d$:

$$J(\pi) = \int \int \frac{\partial \mathcal{L}_d(s, a, d, \log \pi)}{\partial d(s, a, \log \pi)} \, ds \, da = \int \int F(s, a, \pi, \pi') \, ds \, da, \quad (6)$$

where we denote $F = \partial \mathcal{L}_d / \partial d$ and $F$ is assumed to be continuously differentiable with respect to $s, a, \pi$ and $\pi'$ (derivative of $\pi$). To more robustly learn the discriminator $d$, inspired by the idea of adversarial learning (Lowd & Meek, 2005), we make the policy $\pi$ challenge the discriminator $d$ by doing the opposite to the task of $d$ (i.e., minimize $\mathcal{L}_d$), in other words, we want $\pi$ to maximize $\mathcal{L}_d$ under current $d$. By doing so, the policy will provide as little information in $\log \pi$ as possible such that minimizing $\mathcal{L}_d$ becomes harder for the discriminator, this can be seen as minimizing the worst-case error (Carlini et al., 2019; Fawzi et al., 2016; Goodfellow et al., 2014b), which makes the robustness of the discriminator significantly improved.

To maximize $\mathcal{L}_d$ for $\pi$ under current $d$, we let the functional $J(\pi)$ attains its maxima with respect to $\pi$. According to the calculus of variations (Gelfand et al., 2000), the extrema (maxima or minima) of functional $J(\pi)$ can be obtained by finding a function $\pi$ such that the functional derivative of $J(\pi)$ is equal to zero. We can show with theoretical derivation (see Appendix A.2) that ensuring the functional derivative of $J(\pi)$ equal to zero requires $\partial F / \partial \theta_\pi = 0$, where $\theta_\pi$ is the parameters of $\pi$. By the chain rule, we can have

$$\frac{\partial F}{\partial \theta_\pi} = \frac{\partial F}{\partial \log \pi} \cdot \nabla_{\theta_\pi} \log \pi = - \mathop{\mathbb{E}}_{(s,a)\sim\mathcal{D}_e} \left[ \frac{\eta}{d} \cdot \nabla_{\theta_\pi} \log \pi(a|s) \right] +$$
$$\mathop{\mathbb{E}}_{(s,a)\sim\mathcal{D}_o} \left[ \frac{1}{1-d} \cdot \nabla_{\theta_\pi} \log \pi(a|s) \right] - \mathop{\mathbb{E}}_{(s,a)\sim\mathcal{D}_e} \left[ \frac{\eta}{1-d} \cdot \nabla_{\theta_\pi} \log \pi(a|s) \right], \quad (7)$$

where we write the output value of $d(s, a, \log \pi(a|s))$ as $d$ for simplicity by slightly abusing notations. Notice that above derviative can be equivalently perceived as the gradient of a new loss term $\mathcal{L}_w$ of the policy $\pi$ ($\partial \mathcal{L}_w / \partial \theta_\pi = -\partial F / \partial \theta_\pi$) with following form:

$$\mathcal{L}_w = \mathop{\mathbb{E}}_{(s,a)\sim\mathcal{D}_e} \left[ \log \pi(a|s) \cdot \frac{\eta}{d} \right] - \mathop{\mathbb{E}}_{(s,a)\sim\mathcal{D}_o} \left[ \log \pi(a|s) \cdot \frac{1}{1-d} \right] + \mathop{\mathbb{E}}_{(s,a)\sim\mathcal{D}_e} \left[ \log \pi(a|s) \cdot \frac{\eta}{1-d} \right].$$

Hence minimizing $\mathcal{L}_w$ with respect to $\pi$ (make $\partial \mathcal{L}_w / \partial \theta_\pi = 0$) will drive the functional derivative of $J(\pi)$ to zero, which leads to the maxima of $J(\pi)$ (update with gradient direction $-\partial F / \partial \theta_\pi$). Adding this loss term $\mathcal{L}_w$ to the BC task, we get the following new learning objective of $\pi$ as:

$$\min_\pi \alpha \mathop{\mathbb{E}}_{(s,a)\sim\mathcal{D}_e} [-\log \pi(a|s)] - \mathop{\mathbb{E}}_{(s,a)\sim\mathcal{D}_e} \left[ -\log \pi(a|s) \cdot \frac{\eta}{d(1-d)} \right] + \mathop{\mathbb{E}}_{(s,a)\sim\mathcal{D}_o} \left[ -\log \pi(a|s) \cdot \frac{1}{1-d} \right],$$
$$\quad (8)$$

Figure 1: Illustration of DWBC learning mechanism

where $\alpha$ is the weight factor ($\alpha \geq 1$). This new objective essentially transforms the original BC task into a cost-sensitive learning problem (Ling & Sheng, 2008) by imposing the following weight on imitating each state-action transition as

$$\text{Behavioral cloning weights} = \begin{cases} \alpha - \eta/d(1-d), & (s,a) \in \mathcal{D}_e \\ 1/(1-d), & (s,a) \in \mathcal{D}_o \end{cases}. \tag{9}$$

Note that the derivation of the above behavioral cloning weights requires uniform continuity to be satisfied in $F$ and its derivative (details see Appendix A.2). The involvement of $1/d$ and $1/(1-d)$ may violate the continuity assumption. We thus clip the value $d$ to the range of $[0.1, 0.9]$.

Above behavioral cloning weights induce different behaviors on the imitation of transitions from $\mathcal{D}_e$ and $\mathcal{D}_o$. Supposed $d$ is learned to be optimal, i.e., assigns large values (close to 1) to expert transitions and small values to non-expert transitions (close to 0). The weight of those expert transitions in $\mathcal{D}_o$ will become large while the weight of those non-expert transitions will become small. For transitions in $\mathcal{D}_e$, their weights can be adjusted by tuning the parameter $\alpha$. Note that even if the discriminator is learned to be totally wrong (i.e., assign small values to expert transitions and large values to non-expert transitions), which may occur at the very beginning of training, the behavior cloning weights $\alpha - \eta/d(1-d)$ ($\alpha \geq 1, \eta < 1$) will not be drastically changed under value clipping. This means that the policy can still learn from the expert dataset $\mathcal{D}_e$. Even though the weight for $\mathcal{D}_e$ is temporarily incorrect, it will be corrected as the discriminator becomes better and better.

Eq. (8) implies that our approach is also a variant of generalized behavioral cloning objective, as discussed in section 2.2, but uses a different form of weights. Unlike Offline Reinforced Imitation Learning (Zolna et al., 2020a), which uses the discriminator as the reward and learns a value function as the weight, our approach uses the discriminator outputs directly as the weight. This can greatly reduce the training time and avoid the overestimation issue in estimating the value function offline (Kumar et al., 2019). We thus term our algorithm Discriminator-Weighted Behavioral Cloning (DWBC).

A keen reader may find the similarity of our approach to the adversarial learning in GAN (Goodfellow et al., 2014a), which makes the generator (in our case is the policy $\pi$) and the discriminator $d$ learn against each other, thus improves the performance of both tasks. However, the learning strategy of DWBC has several differences compared with GAN and is also much easier to learn. In contrast to the fully adversarial setting in GAN, DWBC adopts a semi-cooperative strategy that shares information between the policy $\pi$ and the discriminator $d$ to improve the performance of both parties. Moreover, GAN needs to solve a min-max optimization problem and is known to suffer from training instability and issues such as mode collapse (Arjovsky et al., 2017). In DWBC, we solve the min-max optimization problem implicitly by transforming the max optimization problem to a new learning loss imposed on the policy. We present an illustration of the learning mechanism of DWBC in Figure 1.

## 4 RELATED WORK

### 4.1 OFFLINE IMITATION LEARNING

Offline IL, which has not received considerable attention, is a promising area because it makes IL more practical to satisfy critical safety desiderata. Offline IL methods can be typically folded into two paradigms: Behavioral Cloning (BC) and Offline Inverse Reinforcement Learning (Offline IRL).

BC (Pomerleau, 1989) is the simplest IL method that can be used in the offline setting, it considers the policy as a conditional distribution $\pi(\cdot|s)$ over actions, recent work (Florence et al., 2021)

enhances BC by using energy-based models (LeCun et al., 2006). BC has shown to have no inferior performance compared to popular IL algorithms such as GAIL (Ho & Ermon, 2016) when clean expert demonstrations are available (Ma, 2020). Unlike BC, offline IRL considers matching the state-action distribution induced by the expert policy, this can be achieved implicitly by adversarial training or explicitly by learning a reward function. Offline IRL algorithms based on adversarial training (Kostrikov et al., 2019; Sun et al., 2021; Swamy et al., 2021; Jarboui & Perchet, 2021) use Intergral Probability Metrics (IPMs) (Sriperumbudur et al., 2009) as a distance measure to solve the dual problem. They introduce a discriminator and aim to find the saddle point of a min-max optimization problem, like GAN (Goodfellow et al., 2014a). Jarrett et al. (2020) avoids the need of min-max problem by fixing the policy to be energy-based models, in such case the KL divergence from the demonstrator's state-action distribution to that of the policy can be computed in closed form. However, recent work finds several fundamental mathematical misconceptions in their proposed approach and we refer the reader to Swamy et al. (2021) for more details.

The common problem of these works is that they imitate equally to all demonstrations, this will hinder the performance if the demonstrations contain suboptimal data. To solve this, Sasaki & Yamashina (2021) reuses another policy learned by BC as the weight of original BC objective, however, this requires that expert data occupies the majority proportion of the offline dataset, otherwise the policy will be misguided to imitate the suboptimal data. Zolna et al. (2020a) and Konyushkova et al. (2020) first construct a reward function that discriminates expert and exploratory trajectories, then use it to solve an offline RL problem. Instead of the adversarial learning scheme, the reward function can also be learned by cascading to two supervised learning steps (Klein et al., 2013). However, offline IRL based on reward learning is extremely expensive to run, requiring solving offline RL in an inner loop, which itself is a challenging problem and prone to training instability (Kumar et al., 2019) and hyperparameter sensitivity (Wu et al., 2019a). Our algorithm can be seen as a combination of these two algorithms in that we use a generalized BC objective to imitate demonstrations selectively and we train a discriminator to distinguish expert and non-expert data and use the output of the discriminator as the weight of the generalized BC objective. There is also one recent work (Chang et al., 2021) that tries to solve the offline IL problem by adopting techniques from pessimistic model-based offline policy learning (Yu et al., 2020; 2021).

## 4.2 OFFLINE REINFORCEMENT LEARNING

One research area highly related to offline IL is offline RL (Lange et al., 2012; Levine et al., 2020), which considers performing effective RL by utilizing arbitrary given, static offline datasets, without any further environment interactions. Note that in offline RL, the training dataset is allowed to have non-optimal trajectories and the reward for each state-action-next state transition triple is known.

Our algorithm draws connection to a branch of methods in offline RL literature that performs "filtered" behavioral cloning explicitly or implicitly. More specifically, Peng et al. (2019), Nair et al. (2020) and Wang et al. (2020) estimate an advantage function, which represents the change in expected return when taking action $a$ instead of following the current policy, and perform weighted regression based on the advantage function, defined as $\mathcal{L}_\pi = \mathbb{E}_{(s,a)\sim\mathcal{D}_b}\left[-\log\pi(a|s)\cdot f\left(A^\pi(s,a)\right)\right]$. The advantage $A^\pi$ can be estimated by Monte-Carlo methods (Schulman et al., 2017; Peng et al., 2019) or Q-value based methods (Schulman et al., 2015; Nair et al., 2020). The filter function $f$ can be a binary filter (Wang et al., 2020) or an exponential filter (Peng et al., 2019; Nair et al., 2020).

While Chen et al. (2021) and Janner et al. (2021) perform filtered behavioral cloning more implicitly. They abstract offline RL as a sequence modeling problem and use Transformer architecture (Vaswani et al., 2017) to perform credit assignments directly via self-attention. Owing to the memorization power of Transformer in capturing long-term dependencies across timesteps, these methods are able to discard low-quality transitions, do behavior cloning only on high-reward transitions across different trajectories and stitch them together to obtain an optimal trajectory.

## 5 EXPERIMENTS

We present empirical evaluations of DWBC in a variety of settings. We start with describing our experimental setup, considered datasets and baseline algorithms. Then we evaluate DWBC against other baseline algorithms on a range of robotic locomotion tasks with different types of datasets.

Finally, we analyze the property of the discriminator. We show that a well-trained discriminator can be used to perform offline policy selection (Fu et al., 2020b), which is of independent interest.

## 5.1 SETTINGS

We construct experiments on both widely-used D4RL MuJoCo datasets and more complex Adroit hand manipulation environment (Fu et al., 2020a). To verify the effectiveness of our methods, we use three setting to generate $\mathcal{D}_e$ and $\mathcal{D}_o$. Note that we use ground truth rewards only to perform this data split step and we discard the rewards afterward.

- In setting 1, we use mixed datasets in Mujoco environments. We sort from high to low of all trajectories based on the total reward summed over the entire trajectory. We define a trajectory as well-performing if it is among the top 20% of all trajectories. We then sample every $X^{\text{th}}$ trajectory from the well-performing trajectories to constitute $\mathcal{D}_e$ and use the remaining trajectories in the dataset to constitute $\mathcal{D}_o$. Note that with $X$ becomes larger, $\mathcal{D}_o$ will contain more proportion of well-performing data.

- In setting 2, we use expert and random datasets in Mujoco environments. We first sample 10 trajectories from expert datasets and 1000 trajectories from random datasets. We then random sample $X$ trajectories from those 10 expert trajectories and combine them with those 1000 random trajectories to constitute $\mathcal{D}_o$, we use the remaining $10 - X$ trajectories to constitute $\mathcal{D}_e$.

- In setting 3, we use human datasets in Adroit environments. We use the same procedure to constitute $\mathcal{D}_e$ and $\mathcal{D}_o$ as in setting 1.

We list all datasets used in this paper and the number of trajectories and transitions in $\mathcal{D}_e$ and $\mathcal{D}_o$ in Appendix C, different $X$ is labeled after the dataset name.

## 5.2 BASELINE AND ABLATED ALGORITHMS

We compare DWBC with the following baseline algorithms:

**BC-exp & BC-all:** Behavioral cloning on expert data or on all data. BC-exp is trained only on $\mathcal{D}_e$, hence it is not exposed to low-performing transitions in $\mathcal{D}_o$. On the other hand, it may not be able to generalize due to the limited size of $\mathcal{D}_e$. BC-all may generalize better than BC-pos due to access to a much larger dataset, but its performance may be negatively impacted by the low-quality data in $\mathcal{D}_o$.

**BCND:** BCND is trained on all data, it reuses another policy learned by BC as the weight of BC, its performance will be worse if the suboptimal data occupies the major part of the offline dataset.

**ORIL:** ORIL learns a reward function and uses it to solve an offline RL problem. It suffers from large computational costs and the difficulty of performing offline RL under distributional shift.

**DWBC-noads:** We include one ablation of DWBC that trains the discriminator without the adversarial learning machinery (i.e., no $\log \pi$ as input), with all others remaining the same as DWBC.

## 5.3 COMPARATIVE EVALUATIONS

We show the comparative results in Table 1 and include the learning curves in Appendix C. It can be shown from Table 1 that DWBC outperforms baseline algorithms on most tasks (25 out of 32 tasks), especially in Mujoco datasets (18 out of 20 tasks), showing that DWBC is well suited to make effective use of the expert dataset $\mathcal{D}_e$ and the mixed quality dataset $\mathcal{D}_o$.

As expected, the performance of BC-exp declines as $X$ becomes larger. This is because that a larger $X$ means the number of well-performing transitions is smaller. In some datasets (e.g., `Halfcheetah_exp-rand-6` and `Ant_mixed-10`), there is no clear winner between BC-exp and BC-all, which suggests that the quality of $\mathcal{D}_o$ for the considered tasks varies. BCND performs poorly compared to other methods due to the majority of low-quality data in the mixed datasets. It usually scores below BC-all. ORIL struggles to learn in some tasks (especially in the `Ant` datasets), which suggests their learned reward function does not accurately contrast expert and

Table 1: Results for Mujoco and Adroit datasets. Scores are undiscounted average returns of the policy at the last iteration of training, averaged over 5 random seeds. We bold the highest values.

| | Task name | BC-exp | BC-all | BCND | ORIL | DWBC-noads | DWBC |
|---|---|---|---|---|---|---|---|
| Hopper | mixed-2 | 1547 | 811 | 437 | 1345 | 2450 | **2531** |
| | mixed-5 | 1263 | 811 | 437 | 998 | 2271 | **2451** |
| | mixed-10 | 1458 | 811 | 437 | 1489 | 1798 | **2231** |
| | exp-rand-3 | 1200 | 314 | 52 | 49 | 1531 | **2231** |
| | exp-rand-6 | 1070 | 314 | 52 | 51 | 1604 | **1610** |
| HalfCheetah | mixed-2 | 4451 | 4210 | 4456 | / | 4980 | **5011** |
| | mixed-5 | 4553 | 4210 | 4456 | 44 | 5011 | **5018** |
| | mixed-10 | 4358 | 4210 | 4456 | 989 | **5022** | 5017 |
| | exp-rand-3 | 6072 | 5753 | 6007 | 6013 | 6021 | **6107** |
| | exp-rand-6 | 5875 | 5753 | 6007 | **6110** | 5803 | 5955 |
| Walker2d | mixed-2 | 2031 | 784 | 760 | 2208 | 2355 | **2436** |
| | mixed-5 | 2014 | 784 | 760 | 2481 | **3112** | 3111 |
| | mixed-10 | 1611 | 784 | 760 | 2384 | 3219 | **3258** |
| | exp-rand-3 | 3078 | 211 | 6 | 955 | 4547 | **4666** |
| | exp-rand-6 | 2871 | 211 | 6 | 5 | 3673 | **4250** |
| Ant | mixed-2 | **2682** | 2255 | 917 | / | 1050 | 2000 |
| | mixed-5 | 2381 | 2255 | 917 | / | 1982 | **3111** |
| | mixed-10 | 2285 | 2255 | 917 | / | 2310 | **3417** |
| | exp-rand-3 | 1071 | 151 | 1045 | 710 | 875 | **1230** |
| | exp-rand-6 | 870 | 151 | 1045 | 639 | 626 | **1127** |
| Pen | human-2 | 1888 | 806 | 1684 | 2262 | **2571** | 2486 |
| | human-3 | 1780 | 806 | 1684 | 2487 | 2362 | **2617** |
| | human-5 | 1531 | 806 | 1684 | 2111 | 2271 | **2487** |
| Door | human-2 | 45 | 31 | -4 | -12 | 51 | **53** |
| | human-3 | **40** | 31 | -4 | -50 | -3 | 10 |
| | human-5 | **38** | 31 | -4 | -3 | 0 | 0 |
| Hammer | human-2 | -187 | -230 | -163 | -222 | **-87** | -88 |
| | human-3 | -191 | -230 | -163 | -237 | -97 | **-96** |
| | human-5 | -213 | -230 | -163 | -159 | -60 | **24** |
| Relocate | human-2 | 4 | 2 | **7** | -4 | 3 | 3 |
| | human-3 | 3 | 2 | **7** | -8 | 0 | 1 |
| | human-5 | 3 | 2 | 7 | **9** | 0 | 1 |

non-expert data. We also find that the performance of ORIL tends to decrease in some tasks (e.g., `Halfcheetah_mixed-5` and `Ant_exp-rand-6`), this "overfitting" phenomenon also occurs in the experiments of offline RL papers (Kumar et al., 2019; Wu et al., 2019a). This is perhaps due to the limited data size and model generalization bottleneck (Neyshabur, 2017).

We also find that DWBC-noads performs worse than DWBC, especially when $\mathcal{D}_o$ contains more expert data, under which circumstance it is hard for the discriminator to distinguish between expert and non-expert data without the help of our proposed adversarial learning machinery.

## 5.4 Additional Experiments

**Offline policy selection by the discriminator.** Offline policy selection (OPS) (Paine et al., 2020; Yang et al., 2020; Dereventsov et al., 2021) considers the problem of choosing the best policy from a set of policies given only offline data. This problem is critical in the offline settings (i.e., offline RL and offline IL) because the online execution is often very costly and safety-aware, deploying a problematic policy may damage the real-world systems (Tang & Wiens, 2021). Note that existing offline RL/IL methods break the offline assumption by evaluating different policies corresponding to their rewards in online environment interactions. However, this online evaluation is often infeasible and hence undermines the initial assumption of offline RL/IL.

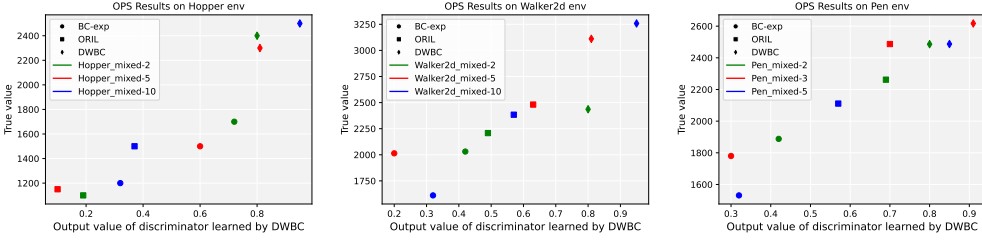

Figure 2: Additional experiment on offline policy selection by the discriminator learned by DWBC.

In this paper, we propose a novel offline policy selection method, simply by using our learned discriminator $d$. More specifically, as the discriminator gets state $s$, action $a$ and the log probability $\log \pi(a|s)$ as input, we can use expert state-action pairs from $\mathcal{D}_e$ and different policy $\pi$ as input. The discriminator learned by DWBC will assign large values (close to 1) when the evaluated policy is close to the policy learned by DWBC, this means that the policy is more optimal.

To validate our proposed idea, we conduct experiments in `Hopper`, `Walker2d` and `Pen` environment. In `Hopper` and `Walker2d`, we use `mixed-2`, `mixed-5` and `mixed-10` datasets, in `Pen`, we use `mixed-2`, `3` and `mixed-5` datasets. We compare three algorithms (BC-exp, ORIL and DWBC) trained in these datasets, total of 9 policies in each environment. We first train DWBC, then we use the learned discriminator along with $\mathcal{D}_e$ to compute the value $d(s, a, \log \pi_i(a|s))$ of each policy $\pi_i$. We plot average $d(s, a, \log \pi_i(a|s))$ versus the policy's true return in Figure 2. As shown, the discriminator's values can well reflect the rank between almost every two policies. This means that we can first train a DWBC policy and then use the trained discriminator $d$ to select the best policy among our candidate policy sets, without executing them in the environment to get the actual returns.

**Comparision of run time.** We also evaluate the run time of training DWBC and other baseline algorithms for 250 thousand training steps (does not include evaluation run time cost). All run time experiments were executed on NVIDIA V100 GPUs. For a fair comparison, we use the same policy network size in BC, BCND, ORIL and DWBC. The discriminator network size is also kept the same in ORIL and DWBC. The results are reported in Figure 3. Unsurprisingly, we find the run time of our approach is only slightly more than BC, while other baselines (ORIL, BCND) are over 7 times more costly than BC. The reason that ORIL is costly to run is due to the additional effort to solve an offline RL problem. The high computation cost of BCND is due to its inner iterations of training $K$ policy ensembles ($K = 5$ in our experiment), which is also mentioned in their paper (Sasaki & Yamashina, 2021). This demonstrates the effectiveness of DWBC by only adds a limited cost to the original BC algorithm while providing substantially improved performance.

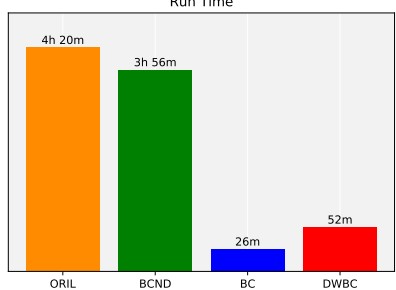

Figure 3: Run time comparison of training each offline IL algorithm.

## 6 CONCLUSION AND LIMITATION

In this paper, we propose a new offline imitation learning algorithm that can learn from suboptimal demonstrations without environment interactions or expert annotations. Experimental results show that our algorithm achieves higher returns and faster training speed compared to baseline algorithms, under different scenarios. One limitation of our work is that since our algorithm is based on weighted BC, the covariate shift problem of BC (Ross et al., 2011) will be inherited. That is, there is no way for the policy to learn how to recover if it deviates from the behavior policy to a new state not seen in the demonstrations. In future work, we will consider modifying the main task from action matching to state-action distribution matching, which is known to be more robust to distributional shift (Kostrikov et al., 2019).

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

# A    DERIVATION DETAILS

In this section, we provide the detailed design intuition and theorectical derivation of DWBC.

## A.1    DECOMPOSITION AND REFORMULATION OF LEARNING TASKS

As discussed in Section 3.1, learning from both the expert dataset $D_e$ and suboptimal dataset $D_o$ implies the need of jointly solving two tasks: a BC task and a discriminator task. A straightfoward solution is to learn the two tasks separately, which solves the BC task by imitating the expert demonstrations in $D_e$ and learn the discriminator via PU learning using data from both $D_e$ and $D_o$:

$$\text{BC task:} \quad \pi(a|s) \leftarrow \arg\min_{\pi} \mathcal{L}_{BC}$$

$$\text{Discriminating task:} \quad d(s,a) \leftarrow \arg\min_{d} \mathcal{L}_d$$

where $\mathcal{L}_{BC}$ and $\mathcal{L}_d$ are discussed and given in objectives (2) and (4) in the main article as follows:

$$\mathcal{L}_{BC} = \mathbb{E}_{(s,a)\sim\mathcal{D}_e}[-\log\pi(a|s)]$$

$$\mathcal{L}_d = \eta \mathbb{E}_{(s,a)\sim\mathcal{D}_e}[-\log d(s,a)] + \mathbb{E}_{(s,a)\sim\mathcal{D}_o}[-\log(1-d(s,a))] - \eta \mathbb{E}_{(s,a)\sim\mathcal{D}_e}[-\log(1-d(s,a))]$$

Naïvely solving above two tasks separately is insufficient. First, the BC task only learn from the expert dataset $D_e$, fails to utilize the potential valuable information in the supoptimal dataset $D_o$. Second, both tasks lack sufficient information to improve their own performance. For example, a good discriminator could provide important information to distinguish the potential expert samples in the suboptimal dataset $D_e$, which are valuable for the learning of policy $\pi$; a well-performed policy $\pi(a|s)$ will assing large probabilities to expert actions under expert states, which could provide additional learning signal for the discriminator $d$ to more easily contrast expert and non-expert transitions in $D_o$.

There are two existing approaches can be used to jointly solve above two tasks, however, both of them have some drawbacks. One approach is to cast the problem into a multi-task style multi-objective optimziation problem, by optimizing an augmented loss $\beta\mathcal{L}_{BC} + (1-\beta)\mathcal{L}_d, \beta \in (0,1)$ for both $\pi$ and $d$. The problem is that the BC task and the discriminating task are different tasks, the potential contradiction of the two tasks in certain settings may impede both tasks from achieving the best perforamnce. Moreover, properly selecting the hyperparameter $\beta$ is very tricky. Another approach is to adopt a GAN-style model (Goodfellow et al., 2014a) which treats the policy as the generator and optimize it implicitly through solving a min-max optimization problem with the discriminator loss $\mathcal{L}_d$. However, this is very costly and is known to suffer from training instability and issues such as mode collapse (Arjovsky et al., 2017). Moreover, although we have an explict loss function $\mathcal{L}_{BC}$ for $\pi$, it is not used in such a GAN-style model, which results in potential loss of information.

In this paper, we design an new cooperative learning mechansim to address above issues, which also results in a computationally efficient practical algorithm. It includes three key ingredients: 1) sharing information between the BC task and the discriminating task to achieve cooperative learning; 2) enabling the BC task to learn on both expert and suboptimal data by introduce an additional corrective loss $\mathcal{L}_w$ impacted by the discriminator outputs; 3) solving both tasks in fully supervised learning manner to maintain computational efficiency. In our approach, we consider following alternative formulation to establish information sharing across the two tasks and enable cooperative learning:

$$\text{BC task:} \quad \pi(a|s) \leftarrow \arg\min_{\pi} \alpha\mathcal{L}_{BC} + \mathcal{L}_w, \quad \alpha > 1$$

$$\text{Discriminating task:} \quad d(s,a,\log\pi(a|s)) \leftarrow \arg\min_{d} \mathcal{L}_d \tag{10}$$

with the new $\mathcal{L}_d$ given in objective (5) as follows:

$$\mathcal{L}_d = \eta \mathbb{E}_{(s,a)\sim\mathcal{D}_e}[-\log d(s,a,\log\pi(a|s))] + \mathbb{E}_{(s,a)\sim\mathcal{D}_o}[-\log(1-d(s,a,\log\pi(a|s)))]$$
$$- \eta \mathbb{E}_{(s,a)\sim\mathcal{D}_e}[-\log(1-d(s,a,\log\pi(a|s)))]$$

In above reformulation, we design the information provided by the policy to discriminator as the element-wise imitation loss value $\log\pi(a|s)$, and the information provided to the policy as the

addional corrective loss term $\mathcal{L}_w$ computed using output values of the discriminator $d$ on samples from both $D_e$ and $D_o$. With the involvement of $\mathcal{L}_w$, we can allow the policy $\pi$ learning from sub-optimal data under the guidance of the discriminator. Moreover, to more robustly learn the discriminator $d$, we borrow the idea of adversairal learning (Lowd & Meek, 2005) and make the policy $\pi$ challenge $d$ by providing as little useful information in $\log \pi(a|s)$ as possible.

We will show in the next section that with the choice of element-wise imitation loss $\log \pi(a|s)$ as the form of information and the advasarial behavior of policy $\pi$, an exact form of $\mathcal{L}_w$ can be derived, and eventually, transforms the original BC task into a cost sensitive learning problem.

### A.2 Drivation of the Corrective Loss Term $\mathcal{L}_w$

In this section, we resort to functional analysis and calculus of variation to derive the exact form of $\mathcal{L}_w$. Under the reformulated problem (10), both the discriminator $d$ and its loss $\mathcal{L}_d$ are impacted by the information provided by policy $\pi$ ($\log \pi(a|s)$). Hence they are now functional of $\pi$. We are interested to see how the variation of $\pi$ impacts $\mathcal{L}_d$, and further influence $d$. Moreover, we can use a specific form of $\mathcal{L}_w$ to alter the behavior of the learned $\pi$ to achieve the desired adversarial behavior.

Denote $\theta_\pi$ and $\theta_d$ as the parameters of the policy $\pi$ and discriminator $d$ respectively. As discussed in Section 3.2, to analyze the impact of $\pi$ on $d$, we fix the parameters $\theta_d$ of $d$ to exclude its own influence. The function of the discriminator $d(\theta_d, s, a, \log \pi(a|s))$ can thus be replaced by $d(s, a, \log \pi(a|s))$. Similarly, with $\theta_d$ fixed, we can define a functional $J(\pi)$ for $\mathcal{L}_d$ which depends on function $\pi$ and further eliminate the effect of changes in $s, a$ on $d$ by integrating over $s$ and $a$ as in Eq.(6):

$$J(\pi) = \int \int \frac{\partial \mathcal{L}_d(s, a, d, \log \pi(a|s))}{\partial d(s, a, \log \pi(a|s))} \, ds \, da = \int \int F(s, a, \pi, \pi') \, ds \, da,$$

where $F = \partial \mathcal{L}_d / \partial d$ is assumed to be a continuously differentiable function with respect to $s, a, \pi$ and $\pi'$ (derivative of $\pi$). Note that as $\theta_d$ is fixed and not considered as a variable in our analysis, $\theta_d$ no longer contributes gradients in the partial derivative $\partial \mathcal{L}_d / \partial d$, and functional $F$ is given as follows:

$$\begin{aligned} F(s, a, \pi, \pi') = - \mathop{\mathbb{E}}_{(s,a) \sim \mathcal{D}_e} \left[ \frac{\eta}{d(s, a, \log \pi(a|s))} \right] + \mathop{\mathbb{E}}_{(s,a) \sim \mathcal{D}_o} \left[ \frac{1}{1 - d(s, a, \log \pi(a|s))} \right] \\ - \mathop{\mathbb{E}}_{(s,a) \sim \mathcal{D}_e} \left[ \frac{\eta}{1 - d(s, a, \log \pi(a|s))} \right], \end{aligned} \tag{11}$$

To enforce the adversarial behavior of $\pi$, we want to make $\pi$ challenge the discriminator $d$ by maximizing $\mathcal{L}_d$ under current $d$ (doing the opposite to $d$). By doing so, the policy will provide as little information in $\log \pi$ as possible such that minimizing $\mathcal{L}_d$ becomes harder for the discriminator. This can be seen as minimizing the worst-case error, which makes the robustness of the discriminator significantly improved.

Maximizing $\mathcal{L}_d$ for $\pi$ under current $d$ is equivalent to finding the maxima of functional $J(\pi)$ ($\mathcal{L}_d$ with $\theta_d$ and $d$ fixed). According to the calculus of variations (Gelfand et al., 2000), the extrema (maxima or minima) of functional $J(\pi)$ can be obtained by finding a function $\pi$ where the functional derivative is equal to zero. We can show with following proposition that ensuring the functional derivative of $J(\pi)$ equal to zero requires $\partial F / \partial \theta_\pi = 0$.

**Proposition 1.** *With the parameters $\theta_d$ of the discriminator $d$ fixed, the functional $J(\pi)$ and $F(s, a, \pi, \pi')$ defined as in Eq.(6) and (11), if uniform continuity of both $F$ and its derivative is satisfied, the necessary condition for $J(\pi)$ attaining its extrema is $\partial F(s, a, \pi, \pi') / \partial \theta_\pi = 0$.*

*Proof.* We derive the finite difference approximation of the functional derivative of $J(\pi)$ by Taylor expansion,

$$J(\pi) = J(\pi_{\theta_\pi^0}) + J'(\pi_{\theta_\pi^0}) \delta \pi + o(\delta \pi), \tag{12}$$

where $\delta \pi$ is the variation of $\pi$. As $o(\delta \pi)$ is a higher order infinitesmall term when $\theta_\pi \to \theta_\pi^0$, which can be ignored. Plug Eq.(12) into the form of $J(\pi)$, we obtain the finite difference approximation of $J(\pi)$ as follows:

$$\Delta J(\pi) = J(\pi) - J(\pi_{\theta_\pi^0}) = J'(\pi_{\theta_\pi^0}) \delta \pi = \frac{d \int \int F(s, a, \pi, \pi') \, ds \, da}{d \theta_\pi} \delta \pi. \tag{13}$$

If the uniform continuity of both $F$ and its derivative is satisfied, we can swap the order of integration and differentiation as well as $ds\,da$ and $\delta\pi$. The above equation can be simplified as:

$$\Delta J(\pi) = \frac{\int\int \partial F(s,a,\pi,\pi')\,ds\,da}{\partial\theta_\pi}\delta\pi = \int\int \frac{\partial F(s,a,\pi,\pi')}{\partial\theta_\pi}\delta\pi\,ds\,da. \tag{14}$$

As $\pi$ can be any function in some function space $\mathcal{F}$. Ensuring the functional derivative of $J(\pi)$ equal to 0 also requires $\partial F(s,a,\pi,\pi')/\partial\theta_\pi = 0$. $\qquad\square$

Given the form of functional $F$ in Eq.(11), we can derive the detailed form of $\partial F/\partial\theta_\pi$ as presented in Eq.(7) of the main article:

$$\frac{\partial F(s,a,\pi,\pi')}{\partial\theta_\pi} = \frac{\partial F(s,a,\pi,\pi')}{\partial\log\pi}\cdot\frac{\partial\log\pi}{\partial\pi}\cdot\frac{\partial\pi}{\partial\theta_\pi} = \frac{\partial F(s,a,\pi,\pi')}{\partial\log\pi}\cdot\nabla_{\theta_\pi}\log\pi$$

$$= -\mathop{\mathbb{E}}_{(s,a)\sim\mathcal{D}_e}\left[\frac{\eta}{d}\cdot\nabla_{\theta_\pi}\log\pi(a|s)\right] + \mathop{\mathbb{E}}_{(s,a)\sim\mathcal{D}_o}\left[\frac{1}{1-d}\cdot\nabla_{\theta_\pi}\log\pi(a|s)\right]$$

$$- \mathop{\mathbb{E}}_{(s,a)\sim\mathcal{D}_e}\left[\frac{\eta}{1-d}\cdot\nabla_{\theta_\pi}\log\pi(a|s)\right],$$

where in the last equation, we slightly abuse the notations and write the output value of $d(s,a,\log\pi(a|s))$ as $d$ for simplicity.

As discussed in Section 3.2, above derivative can be equivalently perceived as the graident of a new loss term for $\pi$, which is exactly the corrective loss term $\mathcal{L}_w$ that we are looking for. If we set $\partial\mathcal{L}_w/\partial\theta_\pi = -\partial F/\partial\theta_\pi$, we recover $\mathcal{L}_w$ in following form:

$$\mathcal{L}_w = \mathop{\mathbb{E}}_{(s,a)\sim\mathcal{D}_e}\left[\log\pi(a|s)\cdot\frac{\eta}{d}\right] - \mathop{\mathbb{E}}_{(s,a)\sim\mathcal{D}_o}\left[\log\pi(a|s)\cdot\frac{1}{1-d}\right] + \mathop{\mathbb{E}}_{(s,a)\sim\mathcal{D}_e}\left[\log\pi(a|s)\cdot\frac{\eta}{1-d}\right]$$

The introduction of the minus sign on $\partial F/\partial\theta_\pi$ is to ensure we find the maxima of $J(\pi)$ rather than its minima. Hence minimizing $\mathcal{L}_w$ with respect to $\pi$ (make $\partial\mathcal{L}_w/\partial\theta_\pi = -\partial F/\partial\theta_\pi = 0$) will drive the functional derivative of $J(\pi)$ to zero, which also lead to the maxima of $J(\pi)$ (update with gradient direction $-\partial F/\partial\theta_\pi$). Adding the new corrective loss term $\mathcal{L}_w$ back to our reformulated problem (10), we obtain the final learning objective of $\pi$ for our BC task (Eq.(8) in the main article):

$$\min_\pi \alpha \mathop{\mathbb{E}}_{(s,a)\sim\mathcal{D}_e}\left[-\log\pi(a|s)\right] - \mathop{\mathbb{E}}_{(s,a)\sim\mathcal{D}_e}\left[-\log\pi(a|s)\cdot\frac{\eta}{d\,(1-d)}\right] + \mathop{\mathbb{E}}_{(s,a)\sim\mathcal{D}_o}\left[-\log\pi(a|s)\cdot\frac{1}{1-d}\right]$$

Note that the derivation of $\mathcal{L}_w$ requires uniform continuity to be satisfied in $F$ and its derivative. The involvement of discriminator output values $1/d(s,a,\log\pi(a|s))$ and $1/(1-d(s,a,\log\pi(a|s)))$ may violate the continuity assumption. We thus clip the discriminator output values to the range of $[0.1, 0.9]$ in our practical algorithm.

## B   TRAINING PROCEDURE DETAILS

### B.1   ALGORITHM DETAILS

In this section, we present the pseudocode of DWBC in Algorithm 1.

### B.2   IMPLEMENTATION DETAILS

In this paper, all experiments are implemented with Tensorflow and executed on NVIDIA V100 GPUs. For all function approximators, we use fully connected neural networks with RELU activations. For policy networks, we use tanh (Gaussian) on outputs. We use Adam for all optimizers. The batch size is 256 and $\gamma$ is 0.99. We rescale the reward to $[0, 1]$ as $r' = (r - r_{\min})\,/\,(r_{\max} - r_{\min})$, where $r_{\max}$ and $r_{\min}$ is the maximum and the minimum reward in the dataset. Note that any affine transformation of the reward function does not change the optimal policy of the MDP. The policy and discriminator network are all 3-layer MLP with 256 hidden units in each layer. The learning rate for the policy is $1e-5$ and the learning rate for the discriminator network is $1e-4$. We search $\alpha$ in $\{1, 2, 5, 10\}$ for best model performance. We clip the output of $d$ to $[0.1, 0.9]$. We set $\eta$ to 0.5 across all tasks, which is the same as the ORIL paper (Zolna et al., 2020a).

---

**Algorithm 1** Discriminator-Weighted Behavior Cloning (DWBC)

---

**Require:** Dataset $D_e$ and $D_o$, hyperparameter $\eta, \alpha$

1: Initialize the imitation policy $\pi$ and the discriminator $d$
2: **while** training **do**
3:     Sample $(s_e, a_e) \sim D_e$ and $(s_o, a_o) \sim D_o$ to form a training batch $\mathcal{B}$
4:     Compute $\log \pi(a|s)$ values for samples in $\mathcal{B}$ using the learned policy $\pi$
5:     Compute discriminator output values $d(s, a, \log \pi(a|s))$ using sampled $(s, a)$ and computed $\log \pi(a|s)$
6:     Update $d$ by minimizing the learning objective in Eq.(5)
7:     Update $\pi$ by minimizing the learning objective in Eq.(8)
8: **end while**

---

## C  ADDITIONAL RESULTS

### C.1  DATASETS DETAILS

In this section, we list all datasets used in our paper and the number of trajectories and transitions in $\mathcal{D}_e$ and $\mathcal{D}_o$, different $X$ is labeled after the dataset name.

| **Dataset-$X$** | #$\mathcal{D}_e$ | | #$\mathcal{D}_o$ | |
|---|---|---|---|---|
| | Trajectories | Transitions | Trajectories | Transitions |
| Hopper_mixed-2 | 204 | 96,222 | 1,835 | 303,737 |
| Hopper_mixed-5 | 82 | 39,590 | 1,957 | 360,369 |
| Hopper_mixed-10 | 41 | 19,176 | 1,998 | 380,783 |
| Hopper_exp-rand-3 | 7 | 6,993 | 1,003 | 23,720 |
| Hopper_exp-rand-6 | 4 | 3,996 | 1,006 | 26,717 |
| Halfcheetah_mixed-2 | 20 | 19,980 | 182 | 181,818 |
| Halfcheetah_mixed-5 | 8 | 7,992 | 194 | 193,806 |
| Halfcheetah_mixed-10 | 4 | 3,996 | 198 | 197,802 |
| Halfcheetah_exp-rand-3 | 7 | 6,993 | 1,003 | 1001,997 |
| Halfcheetah_exp-rand-6 | 4 | 3,996 | 1,006 | 1004,994 |
| Walker2d_mixed-2 | 109 | 74,857 | 984 | 226,050 |
| Walker2d_mixed-5 | 44 | 31,010 | 1,049 | 269,897 |
| Walker2d_mixed-10 | 22 | 15,569 | 1,071 | 285,338 |
| Walker2d_exp-rand-3 | 7 | 6,993 | 1,003 | 21,874 |
| Walker2d_exp-rand-6 | 4 | 3,996 | 1,006 | 24,871 |
| Ant_mixed-2 | 49 | 46,646 | 436 | 254,869 |
| Ant_mixed-5 | 20 | 19,209 | 465 | 282,306 |
| Ant_mixed-10 | 10 | 9,866 | 475 | 29,1649 |
| Ant_exp-rand-3 | 7 | 6,458 | 1,003 | 182,909 |
| Ant_exp-rand-6 | 4 | 3,996 | 1,006 | 185,371 |
| Pen_human-2 | 3 | 597 | 22 | 4,378 |
| Pen_human-3 | 2 | 398 | 23 | 4,577 |
| Pen_human-5 | 1 | 199 | 24 | 4,776 |
| Door_human-2 | 3 | 770 | 22 | 5,934 |
| Door_human-3 | 2 | 479 | 23 | 6,225 |
| Door_human-5 | 1 | 255 | 24 | 6,449 |
| Hammer_human-2 | 3 | 1,485 | 22 | 9,800 |
| Hammer_human-3 | 2 | 844 | 23 | 10,441 |
| Hammer_hunman-5 | 1 | 483 | 24 | 10,802 |
| Relocate_human-2 | 3 | 1,328 | 22 | 8,589 |
| Relocate_human-3 | 2 | 862 | 23 | 9,055 |
| Relocate_human-5 | 1 | 511 | 24 | 9,406 |

Table 2: Dataset details.

## C.2 LEARNING CURVES

In this section, we provide the learning curves of listed algorithms in Section 5. As the learning procedure of BC is quite fast and stable, for more clearly presentation, we plot the results of BC-exp and BC-all as a horizon bar with the shaded area as the standard deviation across different seeds. The average return in $\mathcal{D}_e$ is plotted as the red dashed line in all plots.

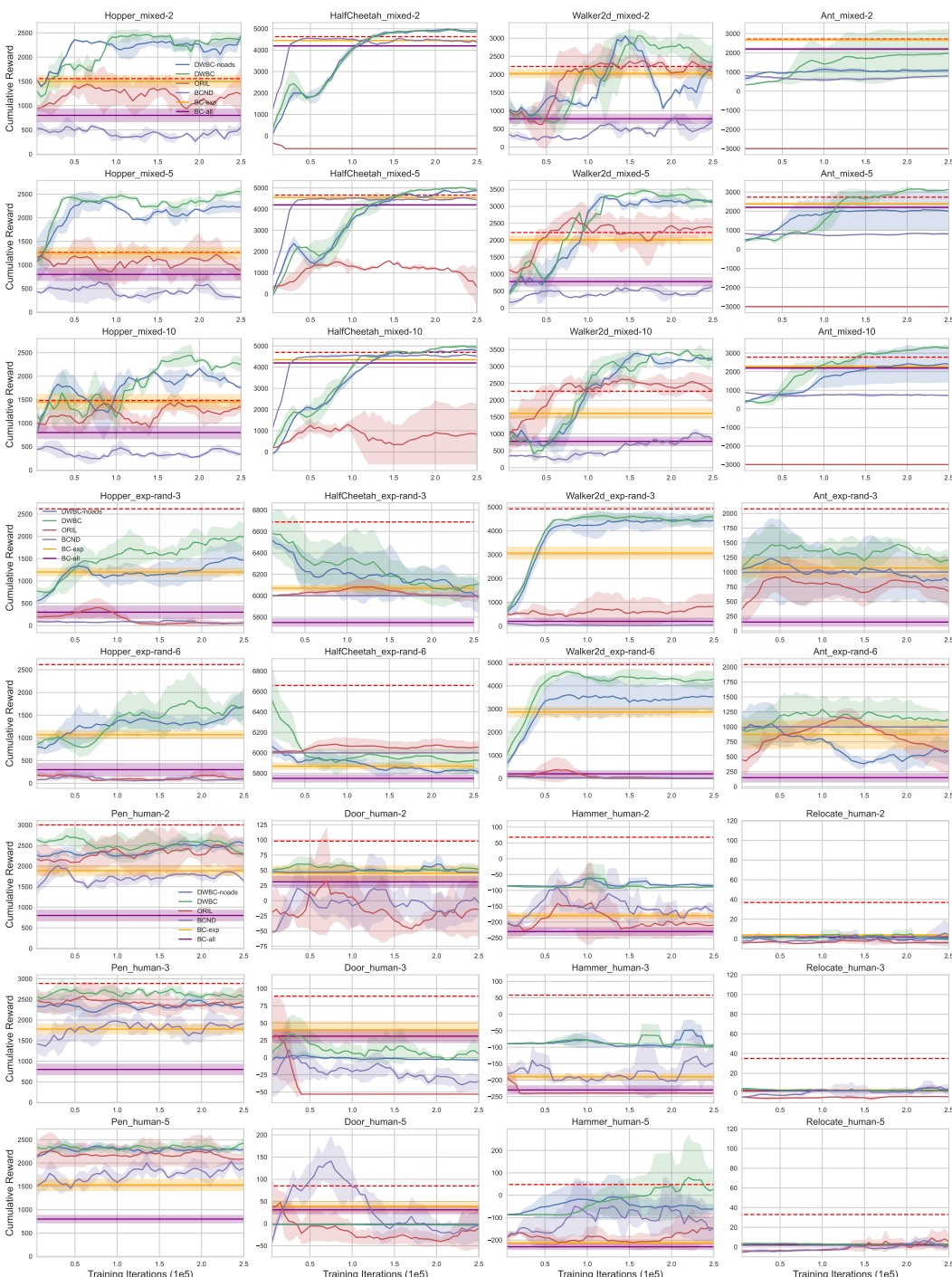

Figure 4: Learning curves of compared algorithms on different datasets.

