# OpenReview forum: "Discriminator-Weighted Offline Imitation Learning from Suboptimal Demonstrations"
_ICLR.cc/2022/Conference — ICLR 2022 Submitted_

### Official Review · Reviewer_nHQm · 2021-10-29

**Correctness:** 2
**Technical Novelty And Significance:** 2
**Empirical Novelty And Significance:** 2
**Recommendation:** 3
**Confidence:** 3

**Main Review:**

**Strengths**
- The paper is overall well written and easy to follow. The justification of the setup makes sense and is well explained.
- The setting is very interesting and has a great potential impact on the community.

**Weaknesses**
- Method
    - The authors fail to mention a very close work that already applied positive-unlabeled learning to the Imitation learning setup: PUGAIL (https://arxiv.org/abs/1911.00459). This greatly limits the novelty of this work.
    - I find the theoretical derivations confusing. In particular, in appendix B, proposition 1, is “when” an “if and only if” condition or just an implication?
    - I also don’t understand why the authors need to introduce the function F. Overall the mathematical derivations are unclear and would benefit for more details (keeping them shorter in the main text but developing them correctly in appendix for example).

- Related Work
    - I think a few more works would be worth mentioning. In particular, some previous work already considered having access to two datasets (on expert, one non-expert), like CSI  (https://hal-supelec.archives-ouvertes.fr/hal-00869804/document)  or MILO (https://arxiv.org/pdf/2106.03207.pdf)
- Evaluation
    - Authors write “the proposed algorithm can learn behaviors that are much closer to the optimal policies” yet they fail at providing experiments that support this claim as they only study the return of the learnt policies. I would recommend that they checkout this work (https://arxiv.org/abs/2105.12034 ) that provides insights on how to evaluate models in the context of imitation learning. What is more, the plots don’t show the average return of D_e which makes it impossible to use the return as a proxy to study how “close” the policy is from the demonstrations.
- Experiments
    - The experimental setup is good but a bit “light”. As written by the authors, their method is quite fast to train, so why not provide more results on different setups, e.g. 1) with very little data (e.g. like 1 or 5 trajectories only)? with very random data in D_o? with human expert data in D_e? with more complicated environments like Adroit? (All these environments/datasets are available in D4RL).
    - Please report the average return in D_e as a horizontal bar in the plots. Otherwise it is impossible to calibrate the results as a reader.
- Writing
    - In the abstract, the authors say “both optimal and non-optimal expert behaviors”. I find this expression a bit confusing. It suggests that the algorithm will only work if the D_o is actually made of expert but slightly suboptimal trajectories.
    - nit: In 3.2, “much hard” -> “much harder”, “which we denote it as” -> “which we denote as”


**Summary Of The Paper:**

The paper deals with the following setup: offline imitation learning in the presence of both an expert dataset and a non-expert dataset. More precisely, the goal is to learn a policy as close as possible to the one(s) that generated the samples in a dataset $D_e$, while making the most of samples in a non-expert dataset $D_o$. The “reward” information is not present/used in the dataset.
The authors draw inspiration from the positive-unlabeled classification as well as the adversarial imitation learning literature to propose a new algorithm to tackle this problem. They interleave the training of a discriminator and a policy. The discriminator is trained to discriminate between expert and non-expert dataset (using a positive/unlabeled loss) and takes as input the state, the action and the logit of the policy $\pi(a | s)$. The policy is trained to imitate the expert on $D_e$ and to “fool” the discriminator.

The authors present results on four environments from the Gym Mujoco suite with datasets extracted from the D4RL datasets.


**Summary Of The Review:**

I believe this is an interesting idea in an interesting setup yet it lacks novelty and it would deserve more work, notably on the experimental part, before being published.

---

> ### Author Response · Authors · 2021-11-21
> **Response to Reviewer nHQm (Part 1/2)**
>
> We thank the reviewer for the thorough and detailed comments. We believe there is a misunderstanding about our method, and we will try to clarify these points in the paper. We address specific points below, but first address what we think is the main misunderstanding.
>
> We want to clarify that using a positive-unlabeled learning objective to optimizes the discriminator is not our main contirbution, as previous works also use similar fashion to construct a better reward function.
> **The contribution of our paper is introducing a novel adversarial scheme to jointly learn the discriminator and the policy.**
> We couple the policy and the discriminator by using the policy's output as part of the discriminator's input. To learn a better discriminator, this scheme introduces an additional corrective loss term to the policy and transforms the original BC task into a cost sensitive learning problem.
> This is **not** like adversarial imitation learning as we are not using the discriminator to distinguish between expert and policy, we use the policy to help the learning of the discriminator. To the best of our knowledge, there is no previous work doing so in a similar fashion.
> Experiments demonstrate that on a set of benchmark domains, the proposed algorithm can leverage the suboptimal dataset to learn more performant policies compared to vanilla behavior cloning objectives and prior offline IL/RL baselines.
>
> >"I find the theoretical derivations confusing. In particular, in appendix B, proposition 1, is “when” an “if and only if” condition or just an implication ?"
>
> * We appologize for the less detailed theoretical derivation in our previous version of the paper. We have revised the vague descrptions in the main text and re-written the appendix to add detailed design intuition and mathematical derivation of DWBC. Please refer to Appendix A in our revised paper for more details.
> * "when" in the Proposition 1 means "necessary condition". We have revised the text of Proposition 1 in Appendix A to remove the ambiguity.
>
> >"I also don’t understand why the authors need to introduce the function $F$."
>
> The derivation of DWBC is based on functional analysis and calculus of variation.
> Due to the involvement of $\log\pi(a|s)$ in the input of the discriminator, $d$ as well as its loss $\mathcal{L}_d$ now become the functionals of $\pi$ (i.e. function of a function). More specifically, $\mathcal{L}_d$ can be considered as $\mathcal{L}_d(\theta_d, \pi)$. To solely analyze the impact of $\pi$ on $\mathcal{L}_d$, we instead define a new functional $J(\pi)$ for $\mathcal{L}_d$ which has $\theta_d$ fixed and no longer considered as a variable. We maximize $J(\pi)$ with respect to $\pi$, and use calculus of variation to identify a necessary condition for the maxima of $J(\pi)$ ($\partial F/\partial \theta_\pi=0$). This is then perceived as minimizing an additional corrective loss term $\mathcal{L}_w$ for the policy $\pi$, and eventually transform the original BC task into a cost sensitive learning problem.
>
> Introducing the functional $F$ and write $J(\pi)$ as an integral form is a commonly used trick in calulus of variation. Such a treatment is also used in deriving the well-known Euler-Lagrangian equation. We write functional $J(\pi)$ in an integral form to eliminate the effect of changes in $s$ and $a$ when dealing with the variation of $\pi$.
> This allows us focusing only on analyzing the impact of function $\pi$ on the functional $J(\pi)$.
>
> >"Overall the mathematical derivations are unclear and would benefit for more details (keeping them shorter in the main text but developing them correctly in appendix for example)."
>
> We thank the reviewer for the suggestion of adding more details in appendix. Please refer to the Appendix A in our revised paper for detail revisions.

---

> > ### Author Response · Authors · 2021-11-21
> > **Response to Reviewer nHQm (Part 2/2)**
> >
> > >About the evaluation
> >
> > We revised “the proposed algorithm can learn behaviors that are much closer to the optimal policies” to "the proposed algorithm can achieve higher returns" in abstract and introduction. This work (https://arxiv.org/abs/2105.12034) includes a diverse set of metrics that can be used to measure the success of a policy in imitating the **expert** demonstrations.
> > However, we argue that in our settings, we are not able to use those metrics because there exists non-expert (i.e., suboptimal) data in our dataset. The optimal policy should neither be close to $\mathcal{D}_{e}$ nor close to $\mathcal{D}_b=\mathcal{D}_e \cup \mathcal{D}_o$. So it doesn't make sense to use those metrics in our experiments.
> >
> > >About the experiments
> >
> > * We added more experiments on different datasets and environments to verify the effectiveness of our methods. More specifically, we added experiments on datasets collected from two completely different policies (one optimal expert policy and another fixed but suboptimal policy), we also added experiments on more complex Adroit environments with human demonstrations (some datasets only contain 1 human trajectory in $\mathcal{D}_{e}$).
> > * We added the average return in $\mathcal{D}_{e}$ as a horizontal bar in all plots.
> >
> > >About the writing
> >
> > We revised this confusing and misleading expressions in abstract and introduction, our experiments show that DWBC can work when $\mathcal{D}_{o}$ consists of random trajectories.

---

> > > ### Comment · Reviewer_nHQm · 2021-11-22
> > > **Answer to authors**
> > >
> > > Thanks for the rebuttal and the revision, in particular for the hard work to integrate all changes in a revision.
> > > I still do have questions.
> > >
> > > I don't understand something fundamental in your answer.
> > > You tell me : "we use the policy to help the learning of the discriminator", yet in the paper it is written: " we make the policy $\pi$ challenge the discriminator $d$ by doing the opposite to the task of $d$".
> > >
> > > To me there is nothing "cooperative" in your training and it is fully adversarial. The discriminator learns to discriminate between two datasets and the policy is trained to fool the discriminator (maximize its loss). Where is anything cooperative here? Why would the discriminator not ignore  $\pi$ if the information is not helping discriminate?
> > >
> > > Concerning the experiments,
> > > How did you compute the expert return (the horizontal bars)? In particular, for the adroit datasets, the numbers you provide are wrong.
> > > Here are the ones I computed:
> > > ```
> > > d4rl_adroit_pen/v1-human
> > > Num trajectories: 25
> > > Average Return: 6326.30517578125
> > > Top 20% average Return: 10944.4365234375
> > > --
> > > d4rl_adroit_relocate/v1-human
> > > Num trajectories: 25
> > > Average Return: 3674.07177734375
> > > Top 20% average Return: 4964.3984375
> > > --
> > > d4rl_adroit_hammer/v1-human
> > > Num trajectories: 25
> > > Average Return: 3071.087158203125
> > > Top 20% average Return: 10297.8251953125
> > > --
> > > d4rl_adroit_door/v1-human
> > > Num trajectories: 25
> > > Average Return: 795.6551513671875
> > > Top 20% average Return: 1327.845947265625
> > > ```
> > > Let me also provide the code that generated this.
> > > ```
> > > import tensorflow_datasets as tfds
> > > import numpy as np
> > >
> > > for dataset_name in ['d4rl_adroit_pen/v1-human', 'd4rl_adroit_relocate/v1-human', 'd4rl_adroit_hammer/v1-human', 'd4rl_adroit_door/v1-human'] :
> > >
> > >   dataset = tfds.load(dataset_name)
> > >   rs = []
> > >   for episode in dataset['train']:
> > >     r = 0
> > >     for step in episode['steps']:
> > >       r += step['reward']
> > >     rs.append(r.numpy())
> > >   print(dataset_name)
> > >   print(f'Num trajectories: {len(rs)}')
> > >   print(f'Average Return: {np.mean(rs)}')
> > >   print(f'Top 20% average Return: {np.mean(np.sort(rs)[-5:])}')
> > >   print('--')
> > > ```
> > >
> > >
> > > What is more, I think the additionnal ablation you provide (DWBC-noads) shows that almost no improvement is done over DWBC. No standard deviation is provided (but as all offline RL algorithms, they are probably big), and the results of DWBC-noads are extremely close to DWBC in every environments. As hyperparameters were selected for DWBC and no tuning was done for the ablation, we can assume they basically perform identically. This enlighten my initial point that the discriminator can ignore a part of its input.

---

> > > > ### Author Response · Authors · 2021-11-25
> > > > **Update (Part 1/2)**
> > > >
> > > > We thank the reviewer for actively participating in the discussion. We really appreciate your time and effort.
> > > >
> > > > >"To me there is nothing "cooperative" in your training and it is fully adversarial. The discriminator learns to discriminate between two datasets and the policy is trained to fool the discriminator (maximize its loss). Where is anything cooperative here? Why would the discriminator not ignore $\pi$ if the information is not helping discriminate?"
> > > >
> > > > Before elaborate on why we called our proposed framework a cooperative framework, let's first discuss why we didn't call our proposed framework an adversarial framework.
> > > >
> > > > **Adversarial means task A and task B are contradictory to each other, an improved performance of one task will lead to a deteriorated performance of another task.**
> > > > For example, in GAIL, the policy aims to generate expert data and the discriminator aims to distinguish between expert data and policy data，if the policy perfectly matches the expert, the discriminator will be unable to distinguish well, and vice versa.
> > > >
> > > > **In contrast to adversarial, cooperative means task A and task B can cooperate with each other to help both tasks, an improved performance of one task will also lead to an improved performance of another task.**
> > > > We can design a novel learning framework as shown in (1). In (1), the policy uses additional information from the discriminator ($\mathcal{L}_w$) for learning, the discriminator also uses additional information from the policy ($\log \pi(a|s)$) for learning. If both $\mathcal{L}_w$ and $\log \pi(a|s)$ can help the learning, (1) is definitely a cooperative learning framework. Note that this is a general framework with different $\mathcal{L}_w$, not necessarily to be DWBC.
> > > >
> > > > $$\begin{equation} \begin{split} \text{BC task:}\quad& \pi(a|s)\leftarrow\arg\min_{\pi} \alpha \mathcal{L}_{BC} + {\color{red}\mathcal{L}_w} \\\\ \text{Discriminating task:}\quad& d(s,a, {\color{red} \log \pi(a|s)})\leftarrow\arg\min_d \mathcal{L}_d \end{split} \end{equation} \tag{1}$$
> > > >
> > > > Here is the tricky part, how can we find a $\mathcal{L}_w$ such that $\mathcal{L}_w$ and $\log \pi(a|s)$ can both help the learning of original tasks? We found that if we let the policy $\pi$ maximize $\mathcal{L}_d$ (in other words, learning opposite to $d$), we can have a such $\mathcal{L}_w$, this $\mathcal{L}_w$ is shown as follows,
> > > >
> > > > $$\begin{equation} \underset{(s, a) \sim \mathcal{D}_e}{\mathbb{E}} \left[-\log \pi(a|s) \cdot \frac{\eta}{d\left(1-d\right)} \right] + \underset{(s, a) \sim \mathcal{D}_o}{\mathbb{E}} \left[-\log \pi(a|s) \cdot \frac{1}{1-d} \right] \end{equation} \tag{2}$$
> > > >
> > > > This $\mathcal{L}_w$ will make the policy imitate the expert data in $\mathcal{D}_o$ according to output of the discriminator.
> > > >
> > > > So now the Domino effect happens, $\mathcal{L}_w$ makes the policy learn better --> a better policy makes the discriminator learn better (please see **"Additional general response"** for why) --> a better discriminator again makes the policy learn better --> a better policy again makes the discriminator learn better...
> > > >
> > > > That's the reason why we called our proposed framework a cooperative framework, rather than an adversarial framework. Although there do contain the adversarial part implicitly, the overall framework is a cooperative framework. To avoid misunderstanding, we are considered to change the expression from “cooperative“ framework to "cooperative yet adversarial" framework, but definitely not "adversarial" framework, we will be happy if you can give some suggestions to us.

---

> > > > > ### Author Response · Authors · 2021-11-25
> > > > > **Update (Part 2/2)**
> > > > >
> > > > > >"Concerning the experiments, How did you compute the expert return (the horizontal bars)? In particular, for the adroit datasets, the numbers you provide are wrong."
> > > > >
> > > > > We really apologize for the wrong calculation. You are right, it is because that the reward re-normlization of our new added experiments is wrong (we falsely use the expert data to do normlization and all data to do re-normlization), this resulting the expert return (red dashed lines) of adroit datasets and exp-rand datasets in mujoco is wrong, we correct them in the following table.
> > > > >
> > > > > |                                               | Average expert return |
> > > > > |  ----                                         | ----  |
> > > > > |  Hopper_exp-rand-3                            | 3617 |
> > > > > |  Hopper_exp-rand-6                            | 3615 |
> > > > > |  HalfCheetah_exp-rand-3                       | 10689 |
> > > > > |  HalfCheetah_exp-rand-6                       | 10659 |
> > > > > |  Walker2d_exp-rand-3                       | 4925 |
> > > > > |  Walker2d_exp-rand-6                       | 4919 |
> > > > > |  Ant_exp-rand-3                       | 4578 |
> > > > > |  Ant_exp-rand-6                       | 5042 |
> > > > > |  Pen_human-2                       | 10914 |
> > > > > |  Pen_human-3                       | 11002 |
> > > > > |  Pen_human-5                       | 11210 |
> > > > > |  Door_human-2                       | 1280 |
> > > > > |  Door_human-3                       | 1358 |
> > > > > |  Door_human-5                       | 1476 |
> > > > > |  Hammer_human-2                       | 10360 |
> > > > > |  Hammer_human-3                       | 12322 |
> > > > > |  Hammer_human-5                       | 16022 |
> > > > > |  Relocate_human-2                       | 4930 |
> > > > > |  Relocate_human-3                       | 5121 |
> > > > > |  Relocate_human-5                       | 5709 |
> > > > >
> > > > > We note that the policy training didn't affect by that because the wrong normalization is done for all algorithms, and reward normalization itself didn't change the optimal policy, we confirm that only the expert return of our new added experiments is wrong, expert return of previous experiments (mixed 2/5/10 datasets in mujoco) is correct.
> > > > >
> > > > > >"What is more, I think the additional ablation you provide (DWBC-noads) shows that almost no improvement is done over DWBC."
> > > > >
> > > > > Please see **"Additional general response"**, DWBC outperform DWBC-noads by at least **10%** on all type of datasets., we also find that DWBC achieves close to **20%** improvement when $\mathcal{D}_o$ contains a large number of expert data.
> > > > >
> > > > >
> > > > > >"As hyperparameters were selected for DWBC and no tuning was done for the ablation."
> > > > >
> > > > > To DWBC after hyperparameter selection, we only remove $\log \pi(s|a)$ in DWBC, while keeping all others the same and re-training DWBC-noads, so everything else remains the same, including the only hyperparameter $\alpha$.

---

### Official Review · Reviewer_jCdL · 2021-10-29

**Correctness:** 3
**Technical Novelty And Significance:** 3
**Empirical Novelty And Significance:** 2
**Recommendation:** 5
**Confidence:** 3

**Main Review:**

The problem studied in this paper is quite relevant and important — the ability to re-use large, noisy offline datasets to solve new tasks. This paper motivates the problem well in the introduction and lays out the preliminaries clearly. The related work section also makes references to much of the relevant work, though the subsection on offline RL could be enhanced by expanding on the limitations of offline RL (something the authors already did in the introduction). The method appears to be novel and the discussion framing this method as a weighted behavior cloning objective connects nicely to prior work. Alongside these strengths, there are a number of concerns which I enumerate below:
1. The derivation presented in section 3.2 is a bit confusing and unintuitive. First, I’m not sure I agree with the justification for the why policy outputs needs to be part of the discriminator input. The authors state that it helps the discriminator distinguish between expert and non-expert actions better, but I would like to see this design choice verified empirically — ie. comparing to a discriminator that only takes the state and action as input, possibly without the adversarial formulation. In addition, the discussion on the adversarial policy learning objective also can benefit from additional motivation and details. In particular, why is “providing as little information in $\log \pi$ as possible” the best way to learn a discriminator? Why is the latter necessarily equivalent to “maximizing $L_d$ for $\pi$”? This subsection is, in my opinion, the weakest part of the paper. The discussion here can be improved by a combination of providing a more formal framework for the adversarial learning problem, complementing the formal framework with intuitive explanations, and connecting the ideas here to prior literature.
2. I am having trouble understanding the derivation details in Appendix B. Specifically, why does $\frac{\partial F}{\partial \log \pi}$ equal to $\frac{\partial L_d}{\partial d} \cdot \frac{\partial d}{\partial \log \pi}$? And also I don’t see where the $\frac{\partial d}{\partial \log \pi}$ term in incorporated into the final derivation at the end of the section. Can you please clarify these steps?
3. One critically missing baseline is to train the discriminator defined in equation (4) once, without all the adversarial learning machinery presented in section 3.2 — and use the discriminator weights to weigh different samples when training the policy. This can help justify all of the additional complexities presented in section 3.2 of updating the discriminator and policy in an alternating optimization scheme.
4. While the proposed method appears to work well in locomotion domains, it remains unclear how the approach would scale to more complex settings, such as the robotic manipulation datasets for the kitchen and adroit tasks in D4RL [1] and the manipulation tasks in robomimic [2]. In principle, it should not be too difficult to run experiments on these datasets as well and such experiments would certainly enhance the scope of this paper. That said, given that ICLR is primarily focused on core machine learning methods and less so on strong empirical evaluations, this is not the primary concern in this review.
5. It is unclear to me why BC-all and BCND are constant lines, while the other baselines are curves. Shouldn’t all the baselines be shown as curves — ie. where the performance is changing across training iterations?
6. While it can be implied from the paper as is, pseudocode or a text description of the full training scheme would be nice to have. In particular, I was wondering how often the discriminator is updated relative to the policy — does one update more frequently than the other?

[1] Fu et al., D4RL: Datasets for Deep Data-Driven Reinforcement Learning, 2020

[2] Mandlekar et al., What Matters in Learning from Offline Human Demonstrations for Robot Manipulation, CoRL 2021

**Summary Of The Paper:**

This paper proposes an offline imitation learning framework that incorporates both optimal and suboptimal datasets to learn decision-making tasks, without requiring any reward annotations. To leverage high reward transitions from the suboptimal dataset, the authors formulate a discriminator that optimizes a positive-unlabeled learning objective, where positive samples come from the optimal dataset and unlabeled samples come from the suboptimal dataset. This discriminator is trained in an adversarial fashion along with the policy, resulting in a behavior cloning objective where samples from the optimal and suboptimal datasets are weighted differently according to the discriminator’s predictions. Experiments demonstrate that on a set of simulated locomotion domains, the proposed algorithm can leverage the suboptimal dataset to learn more performant policies compared to vanilla behavior cloning objectives and prior offline IL/RL baselines.

**Summary Of The Review:**

My reaction to this paper is mixed. On one hand, the introduction, preliminaries, and related work are well laid out. On the other hand, I had several confusions about the method and have some concerns about the experiments (see the main review for specific details). As it stands, I think this paper is marginally below the acceptance threshold. I hope that the authors can diligently address the concerns that I raised, at which point I will reconsider my recommendation.

---

> ### Author Response · Authors · 2021-11-21
> **Response to Reviewer jCdL**
>
> We thank the reviewer for the comment. We have revised the vague descrptions in the main text and re-written the appendix to add detailed design intuition and mathematical derivation of DWBC. Please refer to Appendix A in our revised paper for more details. Regarding the concerns from the reviewer, we describe the discussion as follows:
>
> >"Providing formal framework for design intuition and derivation details."
>
> We've revised the vague descrptions in the main text and re-written the appdendix to provide more information. Specifically, we present the formal formulations of our BC task and discriminating task, as well as our model design intuition in Appendix A.1. In Appendix A.2, we present the detailed derivation of the additional corrective loss term $\mathcal{L}_w$ for $\pi$, which is an artifact for the adversarial behavior of $\pi$. Please refer to our revised paper for more details.
>
> >"Reason why include policy output in the discriminator input." & "Justification of introducing adversarial behavior for $\pi$."
>
> Please refer to our general response for detailed clarification.
>
> >"Empirical verification of discriminator without adversarial $\pi$."
>
> We added the comparision to the baseline that trains the discriminator without the adversarial learning machinery (i.e., no $\log \pi$ as input), the results are shown in Table 1 and the learning curves are provided in Appendix C. It can be shown that without this machinery, the performance of DWBC will drop a lot due to a weakly learned discriminator.
>
> >"2. I am having trouble understanding the derivation details in Appendix B. Specifically, why does $\frac{\partial F}{\partial \log\pi}$ equal to $\frac{\partial L_d}{\partial d}\cdot\frac{\partial d}{\log\pi}$? And also I don’t see where the $\frac{\partial d}{\partial\log\pi}$ term in incorporated into the final derivation at the end of the section. Can you please clarify these steps?"
>
> We apologize for the unclear description about the derivation. The reason that $\frac{\partial F}{\partial \log\pi} = \frac{\partial L_d}{\partial d}\cdot\frac{\partial d}{\partial\log\pi}$ is that the functional $F(s,a,\pi,\pi^\prime)$ is defined as the functional for $\partial \mathcal{L}_d/\partial d$ with the parameters of discriminator $\theta_d$ fixed. As $\mathcal{L}_d$ is a function of discriminator $d$, and $d$ contains $\log\pi$ as an argument, hence by the chain rule, we have $\frac{\partial F}{\partial \log\pi} = \frac{\partial L_d}{\partial d}\cdot\frac{\partial d}{\partial\log\pi}$.
> We have explictly defined the form of $F(s,a,\pi,\pi^\prime)$ in Eq.(11) of Appendix and revised the drivation of the final form of $\partial F/\partial \theta_\pi$ to make them easier to understand. Please check our revised paper for more clear descriptions about the technical details.
>
> >"It is unclear to me why BC-all and BCND are constant lines, while the other baselines are curves."
>
> We included comparasion results in Table 1 and added learning curves of compared algorithms in Appendix C.
>
> >"how often the discriminator is updated relative to the policy — does one update more frequently than the other?"
>
> We added the pseudocode of the full training scheme in Appendex B, we update the discriminator and the policy once per iteration.

---

> > ### Comment · Reviewer_jCdL · 2021-11-22
> > **Reviewer jCdL response to authors**
> >
> > I read the author rebuttal carefully as well as the other reviewers' comments. I see some improvements, notably additional citations and additional empirical evaluations on new domains and a new baseline. I am still concerned about several components of the paper, which I enumerate below:
> > 1. Reviewer nHQm raised a very good point: the authors frame their method as a cooperative optimization objective, yet the policy is working to do optimize precisely the negative of the discriminator objective. The writing suggests a cooperative method, while the derivation suggests an adversarial formulation. Either there is a flaw with the premises of the method, or the authors did not choose appropriate terminology to describe their method. In either case, this is a major issue.
> > 2. In my initial review I made the following comment: "In particular, why is .... Why is the latter necessarily ... The discussion here can be improved by a combination of providing a more formal framework for the adversarial learning problem, complementing the formal framework with intuitive explanations, and connecting the ideas here to prior literature." Having read the revised manuscript, I see very incremental changes to the writing, and by and large my questions here have not been addressed. I was hoping for a more expanded discussion, this would really help to make the motivation for the method more clear.
> > 3. The math still remains confusing. I'm still not understanding the claim that $\frac{\partial F}{\partial \log \pi} = \frac{\partial L_d}{\partial d} \cdot \frac{\partial d}{\partial \log \pi}$. How did you get that? Please let me know via step-by-step atomic mathematical derivations, rather than a verbal explanation. In any case, I still don't see how the term $\frac{\partial d}{\partial \log \pi}$ is incorporated in the final derivation at all. Again, if I am missing something, please let me know via step-by-step mathematical derivations. I don't think the revised manuscript conveys these steps.
> >
> > Given all of these points, I am not convinced the rebuttal has sufficiently addressed my initial concerns.

---

> > > ### Author Response · Authors · 2021-11-25
> > > **Update**
> > >
> > > Thank you for your comments and feedback. We really appreciate your time and effort.
> > >
> > > >"The writing suggests a cooperative method, while the derivation suggests an adversarial formulation. Either there is a flaw with the premises of the method, or the authors did not choose appropriate terminology to describe their method. In either case, this is a major issue."
> > >
> > > Please see response to Reviewer nHQm.
> > >
> > > >"The discussion here can be improved by a combination of providing a more formal framework for the adversarial learning problem, complementing the formal framework with intuitive explanations, and connecting the ideas here to prior literature." Having read the revised manuscript, I see very incremental changes to the writing, and by and large my questions here have not been addressed. I was hoping for a more expanded discussion, this would really help to make the motivation for the method more clear."
> > >
> > > Please also see response to Reviewer nHQm. Actually, we have added intuitive explanations and connecting the ideas to prior literature in Section 3.2, blue parts. We connect our idea to the adversarial learning literature and give the intuition about why we will do so.
> > >
> > > >The math still remains confusing. Please let me know via step-by-step atomic mathematical derivations, rather than a verbal explanation.
> > >
> > > We apologize again for the unclear description about the derivation. You are right, $\frac{\partial F}{\partial \log\pi}$ is not equal to $\frac{\partial L_d}{\partial d}\cdot\frac{\partial d}{\log\pi}$. we have revised that in our paper but we forget to correct it in the previous response.
> > >
> > > Let me now elaborate on that step-by-step.
> > >
> > > First, $\mathcal{L}_d$ is a functional of $\pi$ and $\pi$ wants to maximize $\mathcal{L}_d$ under current $d$. Hence we can define the functional $J(\pi)$ for $\mathcal{L}_d$ as
> > >
> > > $$J(\pi) = \mathcal{L}_d(\pi) = \int\int \frac{\partial \mathcal{L}_d(s,a,d,\log\pi) }{\partial d(s,a,\log\pi)} \ ds \ da,$$
> > > we can exclude $d$ because its parameter $\theta_d$ is fixed.
> > >
> > > We denote $F=\partial \mathcal{L}_d/\partial d$ and we can get
> > >
> > > $$ J(\pi) = \int\int F(s,a,\pi,\pi^{\prime}) \ ds \ da$$
> > >
> > > To maximize $\mathcal{L}_{d}$ for $\pi$, we let the functional $J(\pi)$ attains its maxima with respect to $\pi$. According to the calculus of variations, the extrema (maxima or minima) of functional $J(\pi)$ can be obtained by finding a function $\pi$ such that the functional derivative of $J(\pi)$ is equal to zero.
> > >
> > > To make the functional derivative of $J(\pi)$ equal to zero, we require $\partial F/\partial \theta_\pi=0$.
> > > By the chain rule, we can have
> > >
> > > $$\begin{equation}\begin{split} \frac{\partial F}{\partial\theta_\pi}= \frac{\partial F}{\partial\log\pi} \cdot &\frac{\partial\log\pi}{\partial\theta_\pi} = -\underset{(s, a) \sim \mathcal{D}_e}{\mathbb{E}} \left[\frac{\eta}{d} \cdot \frac{\partial\log\pi(a|s)}{\partial\theta_\pi} \right] +\\\\
> > > &\underset{(s, a) \sim \mathcal{D}_o}{\mathbb{E}} \left[\frac{1}{1-d} \cdot \frac{\partial\log\pi(a|s)}{\partial\theta_\pi} \right] - \underset{(s, a) \sim \mathcal{D}_e}{\mathbb{E}} \left[\frac{\eta}{1-d} \cdot \frac{\partial\log\pi(a|s)}{\partial\theta_\pi} \right] \end{split}\end{equation}$$
> > >
> > > This can be equivalently perceived as the gradient of a new loss term $\mathcal{L}_w$ of the policy $\pi$ ($\partial \mathcal{L}_w/\partial \theta_\pi = - \partial F/\partial \theta_\pi$) with following form
> > >
> > > $$\begin{equation} \mathcal{L}_w = \underset{(s, a) \sim \mathcal{D}_e}{\mathbb{E}} \left[\log \pi(a|s) \cdot \frac{\eta}{d} \right] - \underset{(s, a) \sim \mathcal{D}_o}{\mathbb{E}} \left[\log \pi(a|s) \cdot \frac{1}{1-d} \right] + \underset{(s, a) \sim \mathcal{D}_e}{\mathbb{E}} \left[\log \pi(a|s) \cdot \frac{\eta}{1-d} \right] \end{equation}$$

---

> > > > ### Comment · Reviewer_jCdL · 2021-11-28
> > > > **Followup question**
> > > >
> > > > I would like to thank the authors for the additional clarification.
> > > >
> > > > I'm going to defer my thoughts on the "cooperative" nature of this method to a separate discussion with the reviewers.
> > > >
> > > > I do have one followup question. In the response above, the authors imply the following:
> > > > $$\begin{equation}\begin{split} \frac{\partial F}{\partial\log\pi} = -\underset{(s, a) \sim \mathcal{D}e}{\mathbb{E}} \left[\frac{\eta}{d} \right] +\\
> > > > &\underset{(s, a) \sim \mathcal{D}o}{\mathbb{E}} \left[\frac{1}{1-d} \right] - \underset{(s, a) \sim \mathcal{D}e}{\mathbb{E}} \left[\frac{\eta}{1-d} \right] \end{split}\end{equation}$$
> > > >
> > > > However, isn't the quantity in the right actually supposed to equal to $F$ rather than $\frac{\partial F}{\partial\log\pi}$? That's at least my understanding according to how the authors define $F$ in equation 11.

---

> > > > > ### Author Response · Authors · 2021-11-30
> > > > > **Response to followup question**
> > > > >
> > > > > Thank you for your comments and feedback.
> > > > >
> > > > > We appologize for the lack of clarity in this part. As discussed previously, we need to ensure $\partial F/\partial \theta_\pi = 0$, by the chain rule, we have
> > > > > > $\frac{\partial F}{\partial \theta_\pi}=\frac{\partial F}{\partial d}\cdot \frac{\partial d}{\partial \log\pi}\cdot \frac{\partial \log\pi}{\partial \pi}\cdot \frac{\partial \pi}{\partial \theta_\pi}=\frac{\partial F}{\partial \log\pi}\cdot\nabla_{\theta_\pi}\log\pi=0$
> > > > >
> > > > > Note that direct computing the derivative $\frac{\partial F}{\partial \log\pi} = \frac{\partial F}{\partial d}\cdot \frac{\partial d}{\partial \log\pi}$ involves evaluating the network gradient $\frac{\partial d}{\partial \log\pi}$ of the discriminator with respect to input $\log\pi$, and using it in the final corrective loss term $\mathcal{L}_w$. This is not desirable as it will couple the learning objective of the BC task with the discriminator network parameters and lead to a heavy-weighted algorithm. In order to have a simpler and light-weighted practical algorithm, we go one step further.
> > > > >
> > > > > For simplicity, we write $\log\pi$ as a function $g$, we then have
> > > > > > $\frac{\partial F}{\partial \log\pi}\cdot\nabla_{\theta_\pi}\log\pi= \frac{\partial F}{\partial g}\cdot\nabla_{\theta_\pi}g=0$
> > > > >
> > > > > Note that if we apply functional integration with respect to $g$ on above equation, we have
> > > > > > $C=\int 0 \delta g = \int \frac{\partial F}{\partial g} \cdot \nabla_{\theta_\pi}g \delta g = \int \frac{\partial F}{\partial g}\delta g \cdot \nabla_{\theta_\pi}g = F\cdot \nabla_{\theta_\pi}g = F\cdot\nabla_{\theta_\pi}\log\pi$
> > > > >
> > > > > In the above equation, as $\nabla_{\theta_\pi}g$ is now a function of $\theta_\pi$ rather than a function of $g$ and is also continuous differentiable ($\log\pi$ is continuously differentiable by our model design), hence we can swap the terms in above integration. Moreover, as $\pi$ can be any function in some function space $\mathcal{F}$, we further consider a specific boundary condition assumption that makes $C=0$, this results in $F\cdot\nabla_{\theta_\pi}\log\pi=0$.
> > > > >
> > > > > This treatment can greatly simplify our final modeling framework and result in a very simple and light-weighted learning framework, in which both the policy and the discriminator can pass values rather than gradients to each other.

---

### Official Review · Reviewer_xvWD · 2021-11-01

**Correctness:** 2
**Technical Novelty And Significance:** 2
**Empirical Novelty And Significance:** 3
**Recommendation:** 5
**Confidence:** 3

**Main Review:**

Overall, the results are quite promising, especially the fact that the algorithm can learn to mimic the expert from a really small number of expert demonstrations. For example, Figure 1 shows that it is possible to learn HalfCheetah from just 4 thousand expert samples (=4 episodes?). However, I do feel like that both the method and the derivation need some more clarity, and I also have some concerns regarding the experiments.

The derivation includes several inaccuracies and vague statements. The main innovation behind DWBC is to learn a discriminator and condition it on the policy that is being learned. The conditioning is motivated by the fact that it makes discrimination easier (given the optimal policy, one can discriminate the actions based on their probabilities only). It is then argued (after Equation 6) that learning the disciminator becomes more robust, if at the same time, the policy is optimized by minimizing the amount of useful information it can provide to the discriminator. This observation is the most important contribution of the paper, yet the justification seems insufficient and vague. Further, it is not obvious that having the derivative of the discriminator loss w.r.t the policy equal to zero (Equation 7) will be desirable, and that minimization of the final loss in Equation 8 will in fact lead to that condition to be true. That said, the results do indicate the algorithm performs well, so I'm mostly curious seeing a more rigorous derivation and justification that could help shed some light why the DWBC works well.

Moreover, there are several other inaccuracies that make following the derivation hard. For example:
* The loss $L_d$ should not depend on $s$ and $a$ as it does in Equation 6.
* I think in Equation 8, the gradients need to be stopped from flowing to the discriminator.
* How is the discriminator trained? Equation 8 is only minimized with respect to the policy. Is the same loss used for learning the discriminator?

I also have several smaller comments on the experiments:

There seems to be something not quite right with the shading (standard deviation) in Figure 1. For some of the curves, the shaded region is really narrow compared to how much the curves change between iterations. Can you comment on that? Is the training set  different for each seed?

Please add axis labels to Figure 2.

The datasets used in the experiments have a really particular form as they are collected during online training and thus the expert and the sub-optimal data are highly related. It would be good to see a comparison where the datasets come from two completely different policies: one optimal expert policy, and another fixed but suboptimal policy. This would be a more realistic setup (i.e., if the data comes from policy that is trained online, then why do we need offline learning?).

In the offline policy selection experiment, do you use separate datasets for training the discriminator and evaluation? If not, then perhaps the discriminator is simply memorizing the training data.

What is the return of the expert policies for the experiments in Figure 1?


**Summary Of The Paper:**

The paper proposes a new offline imitation learning algorithm, DWBC, for datasets that combine both optimal and suboptimal demonstrations. The approach is based on a modified behavioral cloning loss that weighs expert and non-expert data based on a learned discriminator. DWBC is compared against prior methods in OpenAI Gym tasks, and it is shown to yield better policies compared to the prior work. As a by-product, the method learns a discriminator that can be used to estimate the relative performance of any policies without rolling them out in the environment.

**Summary Of The Review:**

The results presented in the paper are promising, but there are several issues with the clarity of the derivation and some with the experiments, and thus the paper is not yet of sufficient quality to be published as is.

---

> ### Author Response · Authors · 2021-11-21
> **Response to Reviewer xvWD**
>
> We thank the reviewer for the comment. We have revised the vague descrptions in the main text and re-written the appendix to add detailed design intuition and mathematical derivation of DWBC. Please refer to Appendix A in our revised paper for more details. Regarding the concerns from the reviewer, we describe the discussion as follows:
>
> >"The loss Ld should not depend on s and a as it does in Equation 6."
>
> The reviewer is correct that $\mathcal{L}_d$ does not depend on $s$ and $a$. However, our derivation is mainly conducted on $J(\pi)$, which is the functional for $\mathcal{L}_d$ but with the parameter $\theta_d$ of the discriminator fixed, and only consider function $\pi$ as the variable. As $J(\pi)$ is a functional of $\pi$ and $\pi(a|s)$ depends on $s$ and $a$, following the commonly used trick in calculus of variation, we write functional $J(\pi)$ in an integral form to eliminate the effect of changes in $s$ and $a$ when dealing with the variation of $\pi$. Hence although $s$ and $a$ appears in Eq.(6), they are integrated out and not considered as variables for functional $J(\pi)$.
>
> >"I think in Equation 8, the gradients need to be stopped from flowing to the discriminator."
>
> As discussed previously, there is no gradient flow from the policy $\pi$ to the discriminator $d$ in our model. Both the policy and the discriminator are learned in a fully supervised manner with their own optimization objective (8) and (5) respectively. The information from the policy ($\log\pi(a|s)$) and the discriminator output values $d(s,a,\log\pi(a|s))$ used in the additional loss term $\mathcal{L}_w$ are passed as values. There is no gradient passing between policy $\pi$ and discriminator $d$.
>
> >"How is the discriminator trained? Equation 8 is only minimized with respect to the policy. Is the same loss used for learning the discriminator?"
>
> The discriminator is trained by minimizing objective (5) in the paper. In DWBC, for each training step, we first use policy $\pi$ to compute the information $\log\pi(a|s)$ for the discriminator $d$. The computed $\log\pi(a|s)$ values are then fed as input to the discriminator $d$ as input values. We have added a new section in the appendix (Appendix B) to present the detailed pseudocodes of our algorithm in the revised paper.
>
> >“There seems to be something not quite right with the shading (standard deviation) in Figure 1.”
>
> We use the same training set for each seed, the shaded region represents the std across different seeds.
>
> >“Please add axis labels to Figure 2.”
>
> We have added axis labels to Figure 2.
>
> >"It would be good to see a comparison where the datasets come from two completely different policies: one optimal expert policy, and another fixed but suboptimal policy."
>
> We added experiments on datasets collected from two completely different policies (one optimal expert policy and another fixed but suboptimal policy). In the Mujoco environments (Hopper, HalfCheetah, Walker2d and Ant), we first sample 10 trajectories from expert datasets and 1000 trajectories from random datasets. We then random sample $X$ trajectories from those 10 expert trajectories and combine them with those 1000 random trajectories to constitute $\mathcal{D}_o$, we use the remaining $10-X$ trajectories to constitute $\mathcal{D}_e$. We also added experiments on more complex Adroit environments with human demonstrations. The results are shown in Table 1 and the learning curves are provided in Appendix C.
>
> >"In the offline policy selection experiment, do you use separate datasets for training the discriminator and evaluation?"
>
> Offline policy selection (OPS) considers the problem of choosing the best policy from a set of policies given only offline data.
> We first train DWBC using the offline data, then we use the learned discriminator along with $\mathcal{D}_e$ to compute the value $d(s,a,\log\pi_i(a|s))$ of each policy $\pi_i$ that we want to evaluate.
> We plot average $d(s,a,\log\pi_i(a|s))$ versus the policy's true return in Figure 2. As shown, the discriminator's values can well reflect the rank between almost every two policies. This means that we can first train a DWBC policy and then use the trained discriminator $d$ to select the best policy among our candidate policy sets, without executing them in the environment to get the actual returns.
>
> >"What is the return of the expert policies for the experiments in Figure 1?"
>
> We have added the average return in $\mathcal{D}_{e}$ as a horizontal bar in all plots.

---

> > ### Comment · Reviewer_xvWD · 2021-11-25
> > **Response**
> >
> > I thank the authors for their response and the big effort they put to the revision to clarify many of my comments. I however share the same criticism as Reviewer nHQm and will thus not change the score (below borderline). Particularly, it is not clear why and if conditioning the discriminator on the policy improves performance. Discriminator does not receive any additional information from the policy that is not present in the (s,a) pairs already, as the policy is also a function of (s,a) only. Discriminator can thus learn to ignore the additional input. The additional ablation (DWBC-noads) also supports this view as it shows that conditioning does not bring statistically meaningful benefits over DWBC. I agree that having the policy as an input can help shape the optimization landscape and make learning faster, but that appears as not the main claim of the paper.

---

> > > ### Author Response · Authors · 2021-11-25
> > > **Update**
> > >
> > > Please see response to Reviewer nHQm and additional general response.

---

### Official Review · Reviewer_x4DH · 2021-11-05

**Correctness:** 3
**Technical Novelty And Significance:** 3
**Empirical Novelty And Significance:** 2
**Recommendation:** 6
**Confidence:** 4

**Details Of Ethics Concerns:**

No ethics concern.

**Main Review:**

Strengths:
+ The paper is well written and clearly communicates the motivation and contributions.
+ Sufficient empirical experiments which support the core thesis of the paper. The baselines are selected carefully to provide meaningful comparison to the proposed approach.
+ Using the jointly learned discriminator seems like a novel idea for policy optimization and evaluation.


Weakness:
+ The experiments on very similar datasets (e.x changing the fraction of true positives). While I agree that it is strong experiment, From Figure 1. it is clear that the performance of DWBC is not too sensitive to fraction of positive samples. It would be useful to have similar experiments on datasets collected from a random mix of optimal and suboptimal policies (instead of buffer of single learning policy). For example, mixing expert and random policies, or Adroit environments with human demonstrations.
+ Offline Policy evaluation is evaluated on Hopper-v2 only. It would be useful to have a diversity in environments to evaluate this contribution more objectively.

**Summary Of The Paper:**

The authors consider the problem of offline imitation learning in the presence of suboptimal datasets. In the presence of suboptimal data, classical baselines like behavior cloning suffer performance hits, the drop in performance often correlates positively with increase in number of suboptimal trajectories. In this work, the authors propose a novel learning objective inspired by the min-max formulation in GANs. Particularly the agent learns a discriminator to distinguish between samples from the expert and the suboptimal demonstrator. Building on prior work in cost-sensitive learning, this discriminator is used to reweight loss per sample in the offline buffer.

The proposed algorithm is evaluated on standard offline RL benchmarks, across multiple environments. In many environments, e.g Hopper-v2 style environments, the policy improves against strong imitation learning benchmarks. The discriminator (which takes action probabilities as input) is also evaluated in the context of offline policy evaluation. On Hopper-v2 datasets, the discriminator output is compared with true reward accumulated by multiple policies.

**Summary Of The Review:**

Overall, the paper is well motivated and provides promising results for leveraging suboptimal datasets for effective offline imitation learning. The problem is well motivated and the authors provide some novel insights into better objectives for behavior cloning. While the authors demonstrate some improvement over the provided baselines, I would encourage the authors to consider adding couple of more ablations, particularly across datasets with a mix of human and random trajectories.

---

> ### Author Response · Authors · 2021-11-21
> **Response to Reviewer x4DH**
>
> We thank the reviewer for the thorough and detailed comments.
>
> >"It would be useful to have similar experiments on datasets collected from a random mix of optimal and suboptimal policies (instead of buffer of single learning policy). For example, mixing expert and random policies, or Adroit environments with human demonstrations."
>
> We added experiments on datasets collected from two completely different policies (one optimal expert policy and another fixed but suboptimal policy). In the Mujoco environments (Hopper, HalfCheetah, Walker2d and Ant), we first sample 10 trajectories from expert datasets and 1000 trajectories from random datasets. We then random sample $X$ trajectories from those 10 expert trajectories and combine them with those 1000 random trajectories to constitute $\mathcal{D}_o$, we use the remaining $10-X$ trajectories to constitute $\mathcal{D}_e$. We also added experiments on more complex Adroit environments with human demonstrations. The results are shown in Table 1 and the learning curves are provided in Appendix C.
>
> >"Offline Policy evaluation is evaluated on Hopper-v2 only."
>
> We added more offline policy selection experiments in Mujoco (Walker2d) and Adroit (Pen) environments, the results are shown in Figure 2.

---

### Author Response · Authors · 2021-11-21
**General response to reviewers**

Thank you to the reviewers for their time, comments and for providing concrete suggestions on how to improve the paper.

Our paper aims to propose an effective and light-weighted offline imitation learning algorithm to learn from both optimal and suboptimal datasets, without requiring any reward annotations. To leverage (potential) high reward transitions from the suboptimal dataset, we introduce a discriminator that optimizes a positive-unlabeled learning objective, we want to clarify that this is not our main contirbution, as previous works [1] [2] also use similar fashion to construct a better reward function.
**The contribution of our paper is introducing a novel adversarial scheme to jointly learn the discriminator and the policy.**
We couple the policy and the discriminator by using the policy's output as part of the discriminator's input. To learn a better discriminator, this scheme introduces an additional corrective loss term to the policy and transforms the original BC task into a cost sensitive learning problem.
This is **not** like adversarial imitation learning as we are not using the discriminator to distinguish between expert and policy, we use the policy to help the learning of the discriminator. To the best of our knowledge, there is no previous work doing so in a similar fashion.
Experiments demonstrate that on a set of benchmark domains, the proposed algorithm can leverage the suboptimal dataset to learn more performant policies compared to vanilla behavior cloning objectives and prior offline IL/RL baselines.

**Why policy outputs need to be part of the discriminator input?**:

We include the policy's output as additional information to the discriminator. Supposed $\pi$ is learned to be optimal, i.e., assigns large probabilities to expert actions in expert states, the discriminator will receive additional learning signal. It will be easier for the discriminator to contrast expert and non-expert transitions in $\mathcal{D}_o$, as $\log \pi(a|s)$ will be large if $(s,a)$ are from expert behaviors and small if $(s,a)$ are from non-expert behaviors. Without this information from $\pi$, the discriminator is much harder to learn by only using information from $(s,a)$.

**After including policy outputs to be part of the discriminator input, how to change the behavior of $\pi$ such that $d$ can be better learned?**:

To learn a better discriminator $d$, inspired by the idea of adversarial learning, we make the policy $\pi$ challenge the discriminator $d$ by doing the opposite to the task of $d$ (i.e., minimize $\mathcal{L}_d$), in other words, we want $\pi$ to maximize $\mathcal{L}_d$ under current $d$.
This can be seen as minimizing the worst-case error [3] [4], which makes the robustness of the discriminator significantly improved.

Then the overall derivation logic is as follows. Due to the involvement of $\log\pi(a|s)$ in the input of the discriminator, $d$ as well as its loss $\mathcal{L}_d$ now become the functionals of $\pi$ (i.e. function of a function). More specifically, $\mathcal{L}_d$ can be considered as $\mathcal{L}_d(\theta_d, \pi)$. To solely analyze the impact of $\pi$ on $\mathcal{L}_d$, we instead define a new functional $J(\pi)$ for $\mathcal{L}_d$ which has $\theta_d$ fixed and no longer considered as a variable. We maximize $J(\pi)$ with respect to $\pi$, and use calculus of variation to identify a necessary condition for the maxima of $J(\pi)$ ($\partial F/\partial \theta_\pi=0$). This can then be perceived as minimizing an additional corrective loss term $\mathcal{L}_w$ for the policy $\pi$.

Note that DWBC is different from the typical GAN-style framework, which solves a min-max optimization problem: $\min_{\theta_d}\max_{\pi}\mathcal{L}_d(\theta_d, \pi)$. We argue that our scheme can offer several advantages. First, DWBC allows the decoupled training of the policy $\pi$ and discriminator $d$. They can both learn with their own objectives ((8) and (5) in the paper) in a fully supervised manner, which is very easy to train and computationally cheap. Whereas the GAN-style framework requires to solve the complex and costly min-max optimization problem and often suffers from issues such as traning instability and mode collapse. Moreover, the policy in DWBC imitates the expert behavior explicitly by assigning different weights to different samples, while in a GAN-style framework, the policy $\pi$ is implicitly learned through maximizing $\mathcal{L}_d(\theta_d,\pi)$.


[1] Xu et al., Positive-unlabeled reward learning, 2019

[2] Zolna et al., Offline learning from demonstrations and unlabeled experience, 2021

[3] Goodfellow et al., Explaining and harnessing adversarialexamples, 2014

[4] Carlini et al., On evaluating adversarial robustness, 2019

---

### Author Response · Authors · 2021-11-21
**Revision summary**

We would like to thank the reviewers for their detailed comments. We respond to the individual reviews below. We’ve also updated the paper with a number of modifications to address reviewer suggestions and concerns. Summary of updates:
1. We added design intuition of DWBC in Section 3.2 and detailed mathematical derivation in Appendix A;
2. We added more experiments on different datasets and environments to verify the effectiveness of our methods. More specifically, we added experiments on datasets collected from two completely different policies (one optimal expert policy and another fixed but suboptimal policy), we also added experiments on more complex Adroit environments with human demonstrations;
3. We listed all datasets used in our paper and the number of trajectories and transitions in $\mathcal{D}_e$ and $\mathcal{D}_o$ in Appendix C;
4. We added the comparision to the baseline that trains the discriminator without the adversarial learning machinery (i.e., no $\log \pi$ as input);
5. We added more offline policy selection experiments in Mujoco (Walker2d) and Adroit (Pen) environments;
6. We added the pseudocode of the full training scheme in Appendex B;
7. We added learning curves of compared algorithms in Appendix C;
8. We added the average return in $\mathcal{D}_e$ as a horizontal bar in all plots;
9. We added axis labels to Figure 2;
10. We revised some confusing and misleading expressions in abstract and introduction;
11. We added additional references to the related work section:
* A cascaded supervised learning approach to inverse reinforcement learning https://hal-supelec.archives-ouvertes.fr/hal-00869804/document
* Mitigating Covariate Shift in Imitation Learning via Offline Data Without Great Coverage https://arxiv.org/pdf/2106.03207.pdf

12. We added additional references to Section 3:
* Positive-unlabeled reward learning https://arxiv.org/pdf/1911.00459.pdf
* Combating false negatives in adversarial imitation learning https://arxiv.org/pdf/2002.00412.pdf

---

### Author Response · Authors · 2021-11-25
**Additional general response to reviewers about why $d(s,a,\log\pi(a|s))$ can makes $d$ learn better than $d(s,a)$**

Let's give an illustrative example. Suppose $\mathcal{D}_e$ consists of near-end transitions of expert trajectories, for example, transitions from step 60 to step 100 if expert trajectories are 100 steps. States from these transitions are far away from the initial state, which means that we cannot imitate well given only these transitions. Suppose $\mathcal{D}_o$ contains near-front transitions of expert trajectories and transitions from non-expert trajectories. As states from these transitions are near the initial state, they are probably very similar, actions from these states may also bear some similarity. In this setting, using $(s,a)$ may be insufficient for the discriminator to distinguish expert transitions and non-expert transitions in $\mathcal{D}_o$. What about giving these transitions an additional label (such as 1 to expert and 0 to non-expert)? That is the reason why we put $\log\pi(a|s)$ to the input of $d$, with this additional information，$d$ can better accomplish its job.

We want to highlight that, not only the performance of $d$ will decrease, the performance of $\pi$ will also be affected without $\log\pi(a|s)$ as the input of $d$. This is because the policy $\pi$ will give different weights to transition according to $d$, an ill-performed $\pi$ will in turn affect $d$, this is acting as a negative feedback loop.

Experimental results also support our claim, DWBC-noads performs worse than DWBC (the hyperparameter $\alpha$ we used in DWBC-noads and DWBC is the same). DWBC improve DWBC-noads by a large margin especially when $\mathcal{D}_o$ contains more expert data (mixed-10 datasets or exp-rand-6 datasets), under which circumstance it is harder for the discriminator to distinguish between expert and non-expert data, without the help of $\log\pi(a|s)$.

To more clearly see the comparison of DWBC and DWBC-noads, we first normalize the results presented in our paper to values that lie between 0 and 100 according to D4RL, we then compute the mean value by dataset types, shown as follows,

|                                               | mixed-2 mean | mixed-5 mean | mixed-10 mean | exp-rand-3 mean | exp-rand-6 mean |
|  ----                                         | ----  | ----  | ----  | ----  | ----  |
|  DWBC-noads                                   | 50.5 | 58.8 | 57.8 | 56.4 | 50.3 |
|  DWBC                                         | 57.3 | 67.0 | 67.9 | 62.3 | 61.6 |
|  Improvement (compared to DWBC-noads)      | 13.4% | 13.8% | 17.3% | 10.4% | 22.4% |

It can be seen that DWBC outperform DWBC-noads by at least **10%** on all type of datasets., we also find that DWBC achieves close to **20%** improvement when $\mathcal{D}_o$ contains a large number of expert data.

---

### Decision · Program_Chairs · 2022-01-20

**Decision:**

Reject

**Comment:**

The authors introduce a method for offline imitation learning in the presence of optimal and non-optimal data. In particular, they propose to learn a discriminator that can be then further used to modify the behavior cloning loss which leads to performance improvements over baselines. The reviews mention that the idea is novel and most sections of the paper are well written and self-explanatory. They do point out, however, several flaws such as the clarity of the derivation and  the thoroughness of experimental evaluation. While the paper has significantly improved during the rebuttal, its significant changes warrant another round of reviews. I encourage the authors to continue improving the paper, addressing the reviewers' feedback and resubmitting it as it has a potential to be a strong submission.